# Loss of GCNT2/I-branched glycans enhances melanoma growth and survival

Jenna Geddes Sweeney[1,2], Jennifer Liang[1], Aristotelis Antonopoulos[3], Nicholas Giovannone [1,2], Shuli Kang[4], Tony S. Mondala[4], Steven R. Head[4], Sandra L. King[1], Yoshihiko Tani[5], Danielle Brackett[6], Anne Dell[3], George F. Murphy[2,6], Stuart M. Haslam [3], Hans R. Widlund[1,2] & Charles J. Dimitroff[1,2]

Cancer cells often display altered cell-surface glycans compared to their nontransformed counterparts. However, functional contributions of glycans to cancer initiation and progression remain poorly understood. Here, from expression-based analyses across cancer lineages, we found that melanomas exhibit significant transcriptional changes in glycosylation-related genes. This gene signature revealed that, compared to normal melanocytes, melanomas downregulate I-branching glycosyltransferase, GCNT2, leading to a loss of cell-surface I-branched glycans. We found that GCNT2 inversely correlated with clinical progression and that loss of GCNT2 increased melanoma xenograft growth, promoted colony formation, and enhanced cell survival. Conversely, overexpression of GCNT2 decreased melanoma xenograft growth, inhibited colony formation, and increased cell death. More focused analyses revealed reduced signaling responses of two representative glycoprotein families modified by GCNT2, insulin-like growth factor receptor and integrins. Overall, these studies reveal how subtle changes in glycan structure can regulate several malignancy-associated pathways and alter melanoma signaling, growth, and survival.

[1] Department of Dermatology, Brigham and Women's Hospital, Boston, MA 02115, USA. [2] Harvard Medical School, Boston, MA 02115, USA. [3] Imperial College London, Division of Molecular Biosciences, Faculty of Natural Sciences, Biochemistry Building, London SW7 2AZ, UK. [4] The Scripps Research Institute, La Jolla, CA 92037, USA. [5] Japanese Red Cross Kinki Block Blood Center, 7-5-17 Saito Asagi, Ibaraki-shi, Osaka 567-0085, Japan. [6] Department of Pathology, Brigham and Women's Hospital, Boston, MA 02115, USA. Correspondence and requests for materials should be addressed to C.J.D. (email: cdimitroff@bwh.harvard.edu)

Glycosylation is a common post-translational modification with more than 90% of cell-surface proteins and lipids being glycosylated. The glycome, or complete pattern of glycan modifications of a cell, is assembled by the sequential action of glycan-forming and glycan-degrading enzymes, glycosyltransferases and glycosidases, respectively, within the endoplasmic reticulum (ER) and Golgi apparatus[1–3]. Compared with nucleotides and amino acids, glycans can be linked together in many different ways, thus glycans have vast structural complexity and heterogeneity. The numerous functions of glycans are based on their structural diversity and in most instances glycans "tune" function of a protein rather than turning it on or off. While the importance of glycans for proper protein folding and their structural role in extracellular matrix (ECM) have been extensively studied, it is becoming increasingly clear that glycans are also key contributors in regulating intercellular and intracellular signaling, cell trafficking, host–pathogen interactions, and immune responses[4–6].

In cancer, alterations in protein glycosylation are associated with malignant transformation and tumor progression[1,7,8]. One of the most common tumor-associated glycan modifications is the truncation of serine/threonine O-linked glycans (T- and Tn-antigen). Specifically, truncated O-glycans have been shown to directly induce oncogenic features leading to enhanced growth and invasion in pancreatic cancer, and poor outcomes in numerous other cancers[9,10]. Besides truncated O-glycans, increased glycoprotein sialylation has also been shown to promote tumor growth, escape from apoptosis, resistance to therapy, and extravasation and seeding of circulating cancer cells through increased formation of sialyl Lewis X (sLe$^x$) glycans[11,12]. Moreover, increased size and complexity of asparagine (N-linked) glycans, predominantly via augmented expression or activity of N-acetylglucosaminyltransferase V (Mgat5), leads to protumorigenic galectin-ligand formation, enhanced cell motility and invasion, and increased metastatic potential in several cancers, including melanoma[13–15]. Likewise, loss of N-linked glycosylation or presence of core fucosylation on certain signaling molecules, such as epidermal growth factor receptor (EGFR), neural cell adhesion molecule L1 (L1CAM), melanoma cell adhesion molecule (MCAM), vascular endothelial growth factor receptor 2 (VEGFR2) and integrins have been shown to regulate receptor expression, dimerization, cleavage, lectin binding, and signaling in a variety of cancers[15–22]. Thus, although it is clear that aberrant glycans are present on cancer cells, the regulation of global glycosylation patterns in different cancers, and the functional/mechanistic ability of glycans to modulate tumor growth are largely unknown.

Here, we report that among various cancers, melanomas exhibit significant transcriptional changes in glycosylation-related genes. Compared with normal human epidermal melanocytes (NHEMs), this glycome gene blueprint revealed that the β1,6-N-acetylglucosaminyltransferase, GCNT2, is downregulated in melanomas. This led to a loss of asparagine(N)-linked I-branched glycans and the synthesis of poly-N-acetyllactosamine (i-linear) glycans in melanomas. Functionally, we found that knockdown of GCNT2 significantly enhanced melanoma xenograft growth and three-dimensional colony formation and survival, whereas enforced expression of GCNT2 significantly decreased melanoma xenograft growth, and inhibited three-dimensional colony formation and survival. Analyses of two representative N-glycosylated protein families, insulin-like growth factor-1 receptor (IGF1R) and integrins, revealed that GCNT2/I-branched glycan modifications inhibited IGF-1 and ECM-mediated melanoma cell proliferation, survival, and associated downstream signaling pathways. In all, our studies expand our current understanding of the role of aberrant glycans in melanoma and how changes in glycan structure regulate

different malignancy-associated pathways to alter melanoma cell growth and survival.

## Results

**The I-branching enzyme, GCNT2, is downregulated in melanomas.** Altered protein glycosylation is a common feature of cancer, and is known to contribute to malignant behavior. However, how and to what extent the cellular glycome is involved in driving malignant progression is still poorly defined. To address this question, we compared relative expression levels of 698 glycome-related genes from nine different malignancies (GEO GSE29682)[23] (Fig. 1a). Gene Set Enrichment Analysis[24] revealed that melanomas display significant transcriptional changes in the expression of these glycome-related genes (Fig. 1b, c). This observation led us to hypothesize that melanomas may significantly remodel their glycome to help promote malignancy.

To identify specific glycan changes that may contribute to melanoma progression, we used two independent methodologies. First, using mass spectrometry (MS), we compared global N-linked glycan profiles of NHEMs and two human melanoma cell lines, A375 and G361. Concurrently, we analyzed glycosylation-related gene expression levels between NHEMs and A375 and G361 human melanoma cell lines. We found that the N-glycome of NHEMs and both human melanoma cell lines all contained high mannose, N-acetyllactosamines (LacNAcs) appearing as chains of repeating units of N-acetylglucosamine and galactose (polyLacNAcs), and complex N-glycan structures (Supplementary Figure 1a–d). In depth structural analysis of high mass N-glycans revealed that NHEM polyLacNAcs were further modified with branched LacNAcs, known as I-branches (Fig. 2a), while A375 and G361 melanoma cell polyLacNAcs were typically displayed in long linear chains (i-linear glycans) (Fig. 2b, c and Supplementary Fig. 1a–e). In parallel, analysis of glycome-related gene expression patterns in NHEMs and A375 and G361 human melanoma cells revealed that there were 38 differentially-expressed genes common between both melanoma cell lines and NHEMs (Fig. 2d). Of these, the downregulation of β1,6-N-acetylglucosaminyltransferase, GCNT2, in melanoma cells was of particular interest, as this is the enzyme that catalyzes the transfer of N-acetylglucosamine to galactose residues on polyLacNAcs to form I-branched glycans (Fig. 2e). Furthermore, we found that GCNT2 gene expression levels directly correlated with the presence of cell surface I-branched glycans (Supplementary Fig. 2a, b) and culturing GCNT2/I-branched glycan expressing melanoma cells in the presence of complex N-glycan inhibitor, kifunensine, and glycolipid inhibitor, D,l-threo-1-phenyl-2-hexadecanoylamino-3-pyrrolidino-1-propanol-HCl (PPPP), confirmed that I-branched glycans are present on N-linked glycoproteins as well as on glycolipids in melanoma cells (Supplementary Fig. 2c–i). Furthermore, data mining of publicly available datasets from three independent clinical cohorts[25–27], consistently confirmed lower GCNT2 mRNA levels in clinical melanoma specimens compared to normal melanocytes (Fig. 2f).

**GCNT2 expression is downregulated as melanomas progress.** To further assess the expression of GCNT2 in clinical melanoma samples, we developed a dual immunohistochemical (IHC) staining approach using GCNT2 and melanocyte lineage marker, microphthalmia-associated transcription factor (MITF) antibodies to help distinguish GCNT2 in melanocytes/melanoma cells from other cell types. This staining approach revealed strong characteristic nuclear staining of MITF (red), while GCNT2 (brown) staining was observed primarily as cytosolic punctate vesicular subdomains, consistent with the Golgi localization of GCNT2 (Supplementary Fig. 3a). Staining a panel of clinical

specimens, including normal skin, benign nevi, and primary and metastatic melanomas, revealed that GCNT2 is downregulated as melanomas progress from a normal to a malignant and metastatic

state (Fig. 3a–e and Supplementary Fig. 3b, c). This relationship was confirmed by staining a melanoma tissue microarray (TMA), which showed a significant loss of GCNT2 expression from

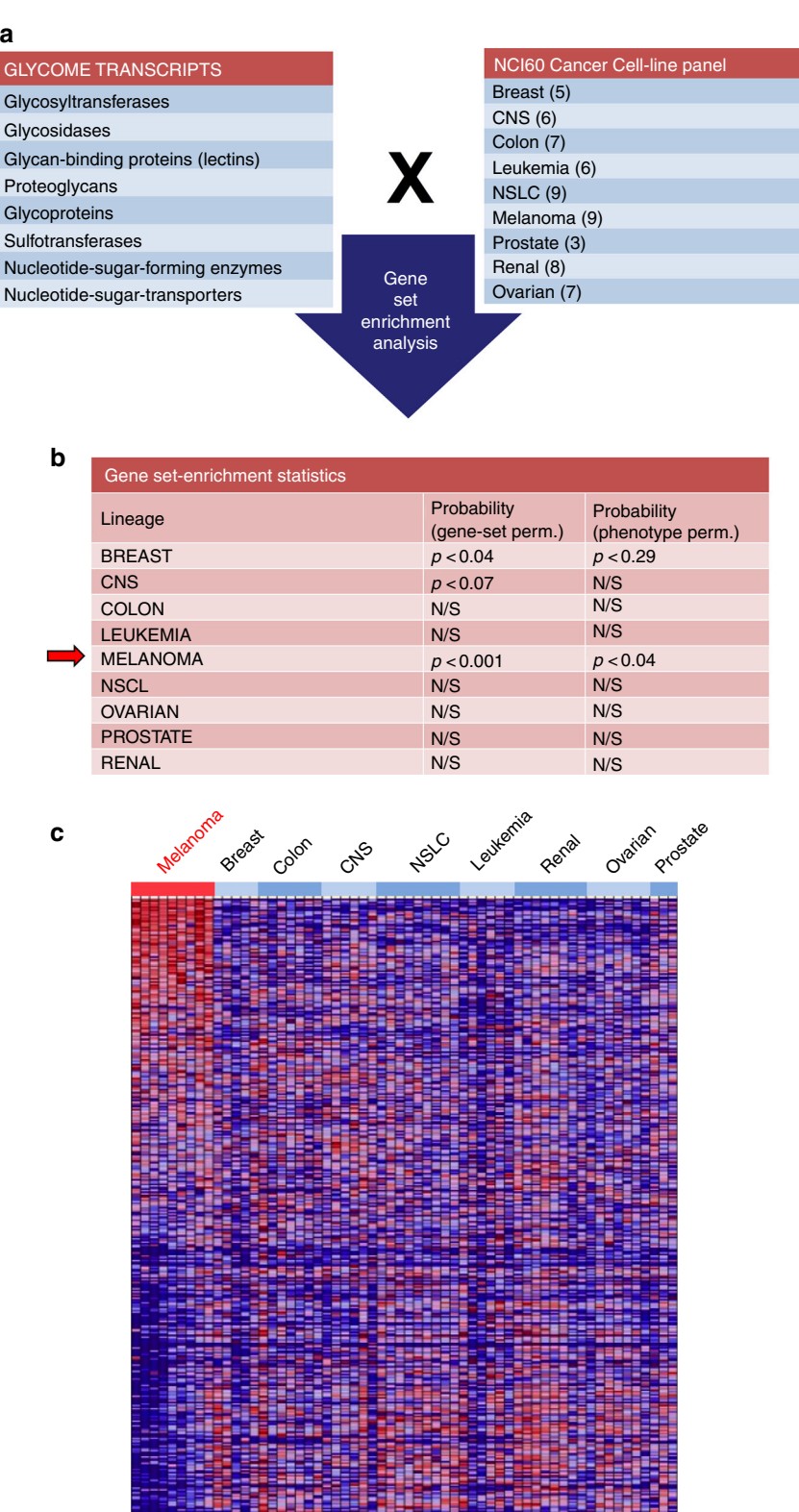

**Fig. 1** Human melanomas are enriched for changes in glycosylation-related genes. **a** Glycome gene transcripts ($n = 698$) selected for gene set enrichment analysis (GSEA) among the NCI-60 cancer cell line panel. **b** Gene- and phenotype-based permutation statistical analysis of GSEA results among the NCI-60 cancer cell line panel. **c** Heat map of glycome gene relative expression (red—increased; blue—decreased) in melanoma cell lines compared with all other cancer cell lines

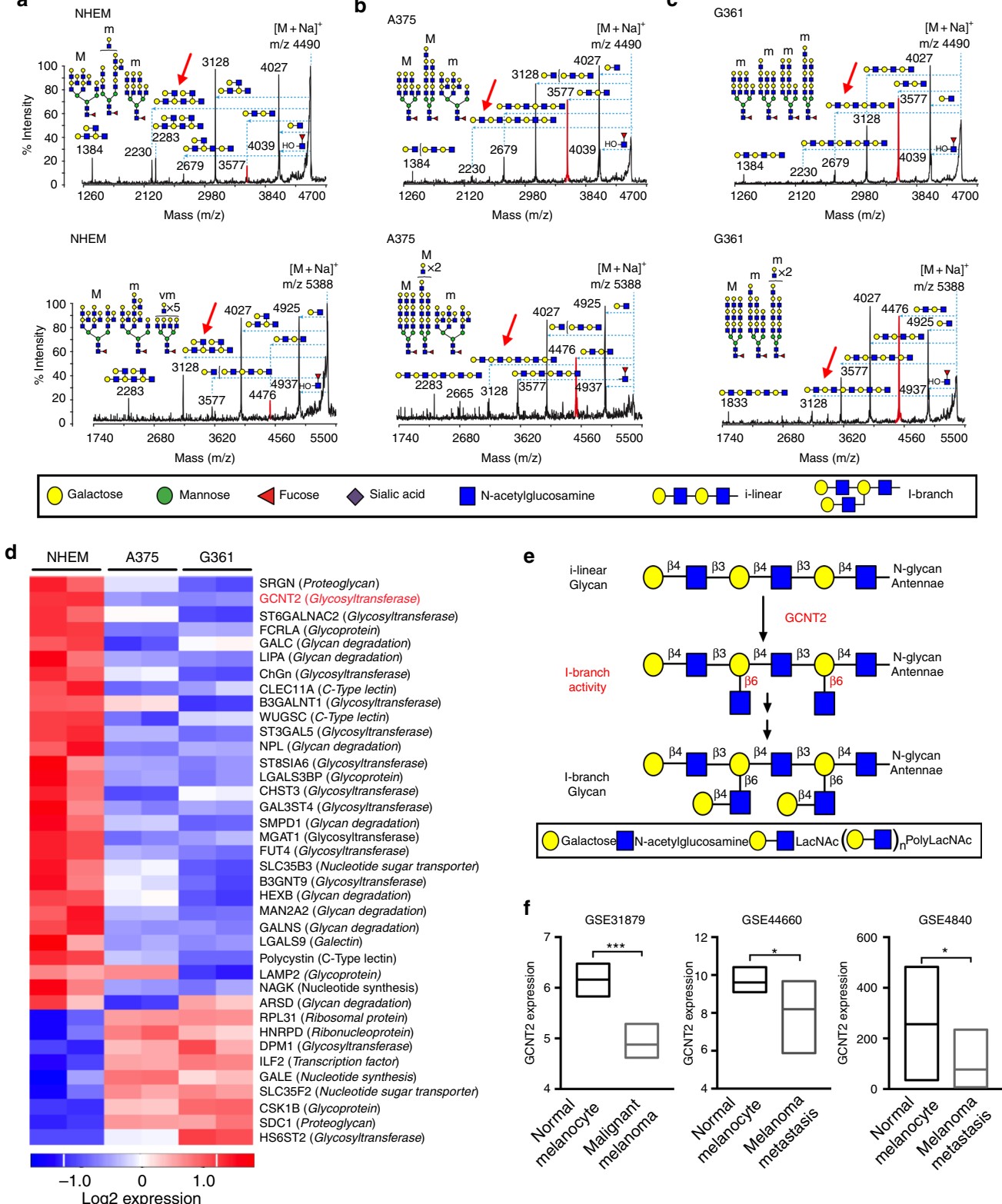

**Fig. 2** Human melanomas downregulate I-branching glycosyltransferase, GCNT2. **a**–**c** MALDI-TOF-TOF MS/MS on high mass spectra from NHEM (**a**), A375 (**b**), and G361 (**c**) human melanoma cells. All parent spectra and ionized subspectra were graphed as % relative intensity and, for clarity, only major ions ([M + Na]+) were shown. Cartoon structures were drawn according to http://www.functionalglycomics.org guidelines. Experiments were repeated on (2) different biological replicates. **d** Heatmap of differentially expressed glycosylation gene transcripts between NHEMs and A375 or G361 human melanoma cells. Statistical analysis was done with limma, and Benjamini–Hochberg correction was used to adjust the p values. Only genes with absolute fold change > 2 and adjusted p value < 0.05 were considered as differentially expressed. **e** Schematic representation of GCNT2 I-branching activity. **f** Box plots illustrating downregulation of GCNT2 in clinical melanoma specimens compared to normal melanocytes in multiple datasets. Statistical analysis was done using unpaired two-tailed Student's t test. Mean is representative line inside the box. See also Supplementary Figs. 1, 2

primary to metastatic melanomas[28] (Fig. 3f). In addition, data mining further confirmed lower GCNT2 mRNA levels in clinical metastatic melanoma specimens compared to primary melanoma specimens from two of three independent clinical cohorts[29–31] (Fig. 3g). Taken together, these data indicate that GCNT2 expression is inversely correlated with melanoma progression and

highlights its potential use as a biomarker that correlates with stage of disease and associated virulence.

**Loss of GCNT2 increases melanoma growth and survival.** Altered glycosylation patterns on cancer cells promote several protumorigenic functions, including enhancing tumor cell

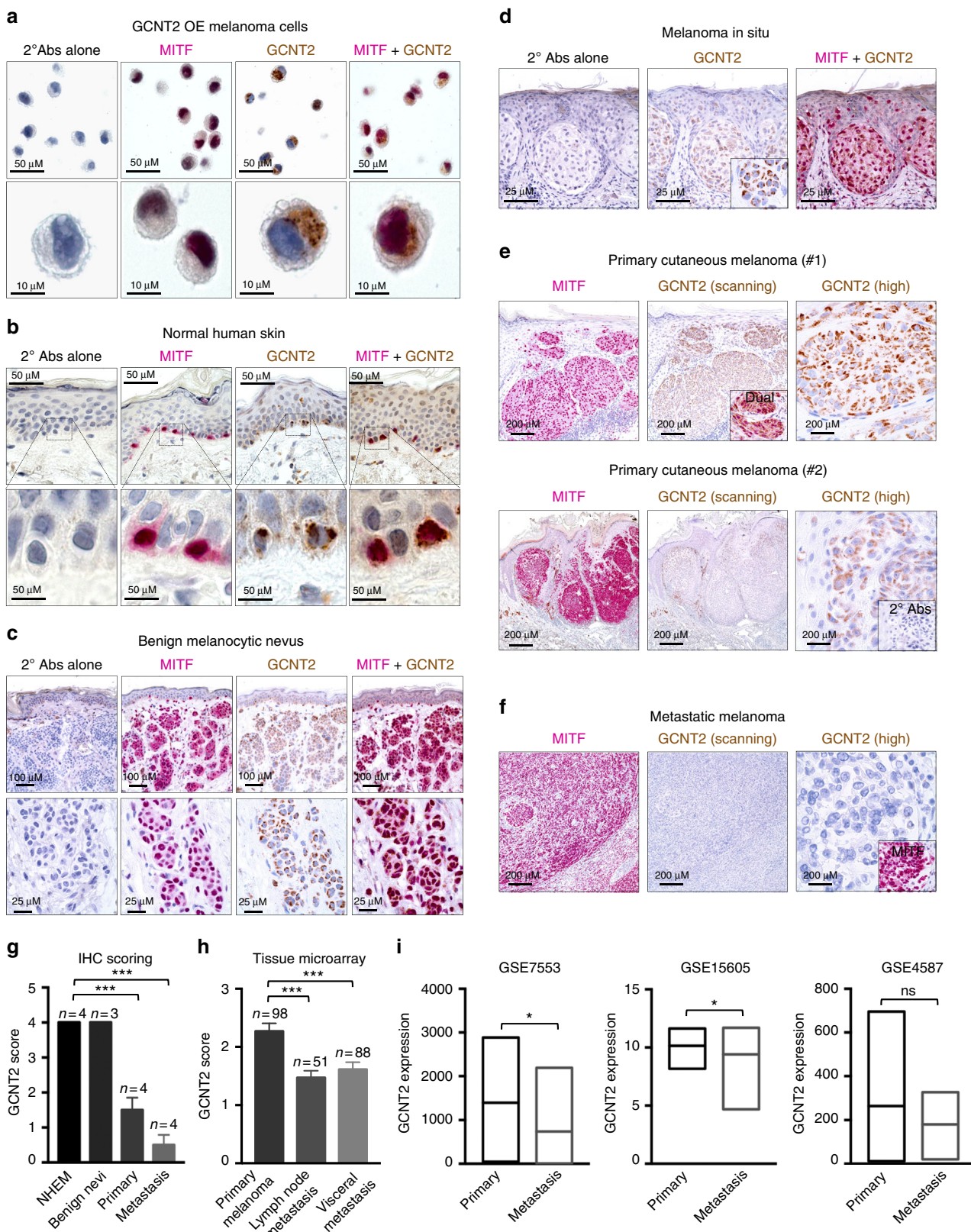

proliferation and survival[1,7]. Therefore, to assess why melanomas decrease expression of GCNT2/I-branched glycans, we generated stable GCNT2 knockdown (KD) and GCNT2 overexpressing (OE) human melanoma cell lines. We confirmed KD and OE of GCNT2/I-branched glycans by immunohistochemistry, qRT-PCR and flow cytometry (Supplementary Fig. 3d, e and Supplementary Fig. 4a–f). We further characterized the cell-surface glycosylation patterns of GCNT2 KD and GCNT2 OE cell variants using a panel of plant lectins known to recognize different glycan structures. Interestingly, we did not observe a significant difference in either α2,6 (SNA lectin) or α2,3 (MAL-II lectin) sialic acid content in GCNT2 KD, GCNT2 OE, or control melanoma cell lines (Supplementary Fig. 4g, h). We also did not observe a significant difference in complex N-glycans (PHA-L lectin) (Supplementary Fig. 4i), suggesting that GCNT2 does not compete with the glycosyltransferases responsible for creating complex N-glycans. However, the presence of GCNT2/I-branched glycans correlated strongly with the binding of polyLacNAc recognizing lectins, STA and LEA (Supplementary Fig. 4j,k), indicating that these lectins favor binding to I-branched poly-LacNAcs over i-linear polyLacNAcs.

Next, we analyzed the role of GCNT2/I-branched glycans on melanoma growth and survival. Functionally, KD of GCNT2 and loss of I-branched glycans increased melanoma tumor xenograft growth and tumor mass compared to control cells (Fig. 4a, b). Conversely, overexpression of GCNT2/I-branched glycans decreased melanoma tumor xenograft growth and tumor mass compared to control cells (Fig. 4c, d). Furthermore, utilizing an anchorage-independent growth assay designed to study tumor-igenicity and survival of malignant cells[32,33], we found that melanoma cells expressing GCNT2/I-branched glycans formed smaller and fewer colonies (Fig. 4e, f). This corresponded to an increase in cell death as measured by flow cytometry (Fig. 4g, h). In addition, GCNT2/I-branched glycan expressing melanoma cells displayed higher levels of proapoptotic genes, BID and BAX, and had lower expression of prosurvival genes, BCL-2 and BCL-XL (Fig. 4i, j). Together, these data indicate that loss of GCNT2/I-branched glycans enhances melanoma growth and survival.

**GCNT2 modifies IGF-1 and ECM growth and survival pathways.** Since glycosyltransferases are global regulators of protein glycosylation, we hypothesized that loss of GCNT2 and its effects on melanoma growth and survival likely arises through modification of numerous glycoprotein targets. Growth factor receptors and integrins are two major classes of cell-surface glycoproteins known to regulate cell growth and survival[34–36]. Furthermore, N-linked glycosylation of EGFR is essential for receptor expression and dimerization[16,17,36], and glycosylation of integrins has been described as a major mechanism regulating integrin processing and activation during cell–cell and cell–ECM interactions[19,35,37]. Thus, we sought to determine whether

GCNT2/I-branched glycans regulate growth factor receptor and/or integrin-mediated melanoma cell growth and survival.

Expression-based analyses using publicly available databases revealed that melanomas have high expression of IGF1R (Supplementary Fig. 5a), as well as high expression of several different integrin α and β chains (Supplementary Fig. 5a). Both IGF1R and all integrins tested were N-glycosylated, as observed by significant shifts in mobility by sodium dodecyl sulphate-polyacrylamide gel electrophoresis (SDS-PAGE)/Western blot analysis following N-glycan digestion with peptide N-glycosidase F (Supplementary Fig. 5b, c). Furthermore, IGF1R and several integrin subunits isolated from melanoma cells expressing GCNT2 exhibited higher LEA lectin reactivity than controls, indicating that IGF1R and α4, β1, and β3 integrins are modified with I-branched glycans (Fig. 5a–d). With this observation, we sought to determine whether differential GCNT2/I-branched glycan expression regulates melanoma growth and survival, in part, through IGF1R and/or through integrin-mediated pathways. Treatment of GCNT2 KD, GCNT2 OE and control melanoma cell variants with IGF-1, the cognate ligand for IGF1R (Fig. 5e), revealed that low GCNT2/I-branched glycan levels increased melanoma cell proliferation (Fig. 5f). Importantly, melanoma cells do not express IGF-1[38], and there was no significant difference in proliferation in control treated cells (Fig. 5f), suggesting that differential GCNT2/I-branched glycan expression can modulate melanoma cell proliferation in an IGF-1-mediated manner.

Because melanoma cells express numerous integrin α and β chains and integrin-ECM interactions are known to regulate cancer cell survival[19], we plated GCNT2 KD, GCNT2 OE, and control cell variants on a mixture of fibronectin, laminin and collagen, common components of the tumor microenvironment and well-described integrin ligands[39] (Fig. 5g). We found that, when plated on ECM, KD of GCNT2/I-branched glycans led to a decrease in proapoptotic Bid protein and gene expression (Fig. 5h, i), and an increase in prosurvival BCL-XL protein and gene expression (Fig. 5h, i). Conversely, overexpression of GCNT2/I-branched glycans led to an increase in proapoptotic Bax protein and gene expression (Fig. 5h, i), and a decrease in prosurvival BCL-XL protein and gene expression (Fig. 5h, i). Of note, although GCNT2 KD melanoma cells plated on plastic did show a slight increase in expression of prosurvival genes BCL-2 and BCL-XL (Supplementary Fig. 5d), upon engagement with ECM, GCNT2 KD melanoma cells significantly upregulated the expression of both BCL-2 and BCL-XL (Fig. 5i). In addition, there was no difference in expression of the proapoptotic genes tested (Supplementary Fig. 5d). Furthermore, GCNT2 OE melanoma cells plated on plastic did not show a significant difference in expression of any of the prosurvival or proapoptotic genes analyzed (Supplementary Fig. 5e), indicating that GCNT2/I-branched glycans decrease melanoma cell survival gene expression in an ECM-dependent manner. Altogether, these data

**Fig. 3** GCNT2 is downregulated as melanomas progress. **a–d** Dual immunohistochemistry (IHC) of GCNT2 overexpressing (OE) melanoma cells (**a**), normal human skin (**b**), benign melanocytic nevus (**c**), melanoma in situ (**d**) and primary and metastatic melanoma (**e**, **f**) clinical specimens stained for MITF and GCNT2. Photomicrographs were taken at 10× (scanning) or at 40× (high). Scale bars are 10, 25, 50, 100, or 200 μM. **g** Semiquantitative cell staining analysis of GCNT2 expression. Percentage of GCNT2[+] cells, which were also MITF[+], was enumerated in normal skin (>50 MITF[+] cells) and in tumor tissues (>100 MITF[+] cells). GCNT2 scoring: 0–1 (1–25% cells positive), 2 (25–50% cells positive), 3 (50–75% cells positive), and 4 (75–100% cells positive). Results representative of 3–4 independent specimens (***$p < 0.001$). **h** Quantitation of GCNT2 staining in tissue microarray containing human cutaneous thick melanomas ($n = 98$) and metastatic melanomas ($n = 139$) cores. GCNT2 scoring: 0–1 (1–25% cells positive), 2 (25–50% cells positive), 3 (50–75% cells positive), and 4 (75–100% cells positive), (***$p < 0.001$). **i** Box plots illustrating downregulation of GCNT2 in metastatic melanoma clinical specimens compared to primary melanoma clinical specimens in multiple datasets. Mean is representative line inside the box (*$p < 0.05$). Statistical analysis was done using unpaired two-tailed Student's $t$ test. See also Supplementary Fig. 3

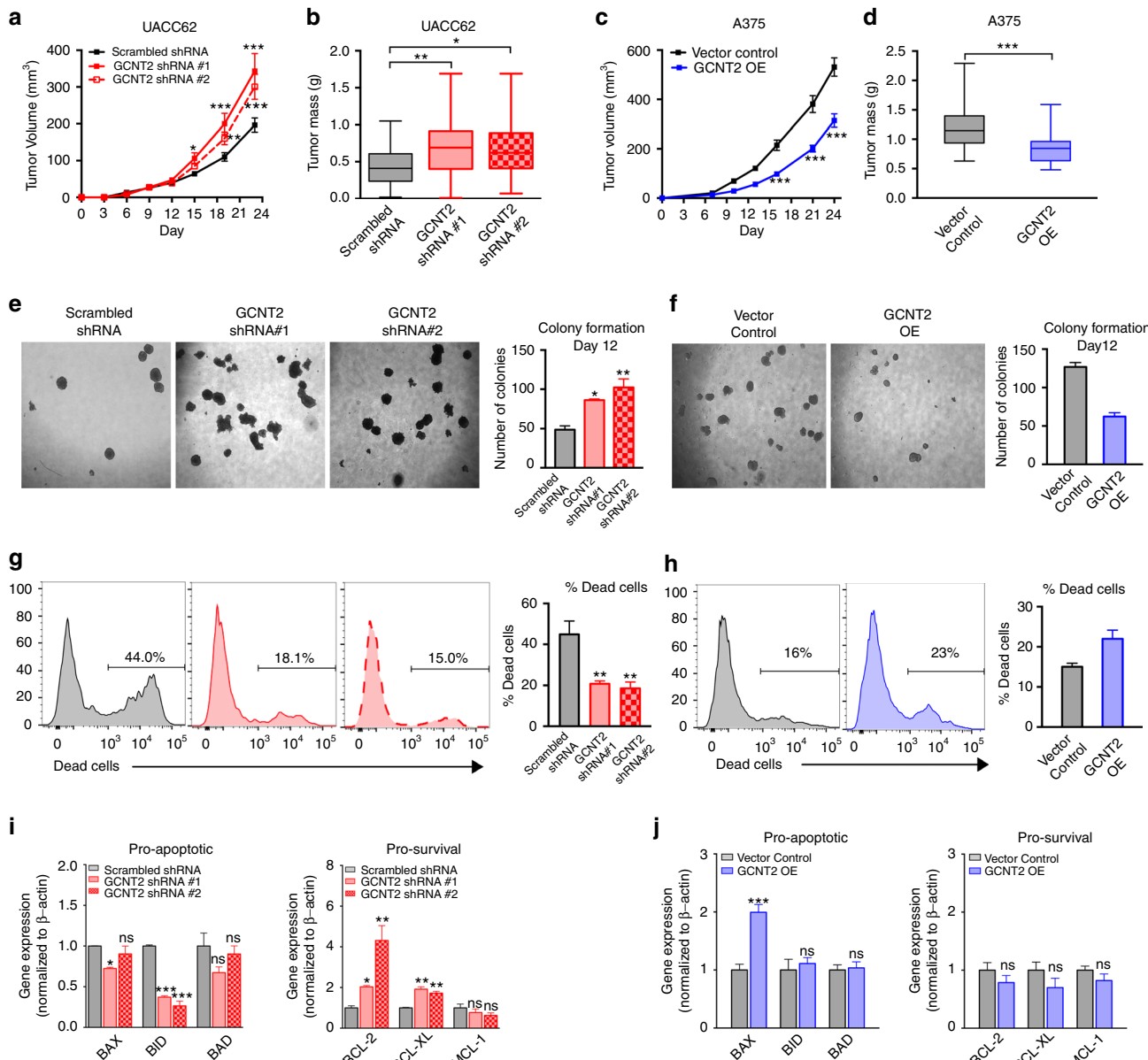

**Fig. 4** Loss of GCNT2 enhances melanoma growth and survival. **a**, **b** Tumor growth kinetics (**a**) and mass (**b**) of UACC62 scrambled control and GCNT2 KD cell variants in NSG mice. **c**, **d** Tumor growth kinetics (**c**) and mass (**d**) of A375 vector control and GCNT2 OE cell variants in NSG mice. **e**, **f** Photographs (left) and quantitation (right) of UACC62 scrambled control and GCNT2 KD (**e**) and A375 vector control and GCNT2 OE (**f**) single cell colony formation in 1% methylcellulose. **g**, **h** Representative flow cytometry plots (left) and quantitation (right) of dead cells from UACC62 scrambled control and GCNT2 KD cell colonies (**g**) and A375 vector control and GCNT2 OE (**h**) cell colonies. **i**, **j** Apoptotic/survival family gene expression measured by qRT-PCR in UACC62 scrambled control and GCNT2 KD (**i**) cell colonies and A375 vector control and GCNT2 OE (**j**) cell colonies. Statistical analyses were done using one-way ANOVA with Dunnett's multiple comparisons test, two-way ANOVA with Dunnett's multiple comparisons test, or unpaired two-tailed Student's $t$ test. Results are representative of $n = 20$ tumors from $n = 4$ experiments and $n = 6$ methylcellulose wells from $n = 2$ experiments (mean ± SEM; $^{**}p < 0.01$; $^{***}p < 0.001$). See also Supplementary Fig. 4

suggest that GCNT2/I-branched glycans regulate IGF1R and integrin growth and survival pathways in melanoma cells.

**Loss of GCNT2 enhances IGF1R and integrin-ECM signaling.**
To dissect how differential GCNT2/I-branched glycan expression modulates IGF1R and integrin activities in melanoma cells, we analyzed related signaling events. Treatment of GCNT2 KD, GCNT2 OE and control cell variants with IGF-1 (Fig. 6a), revealed that KD of GCNT2 and I-branched glycans increased IGF1R tyrosine phosphorylation and AKT serine and threonine

phosphorylation (Fig. 6b, c), a downstream target and signaling molecule known to be involved in cell proliferation and survival[40]. Conversely, overexpression of GCNT2/I-branched glycans decreased IGF1R tyrosine phosphorylation, as well as downstream AKT serine and threonine phosphorylation (Fig. 6b, c). Importantly, there was no significant difference in either gene or cell-surface protein expression of IGF1R in GCNT2 KD and GCNT2 OE cell variants compared to control cells (Supplementary Fig. 6a–d), indicating that cell-surface expression differences between GCNT2 KD and GCNT2 OE cells and their controls cannot account for altered signaling. These data indicate that

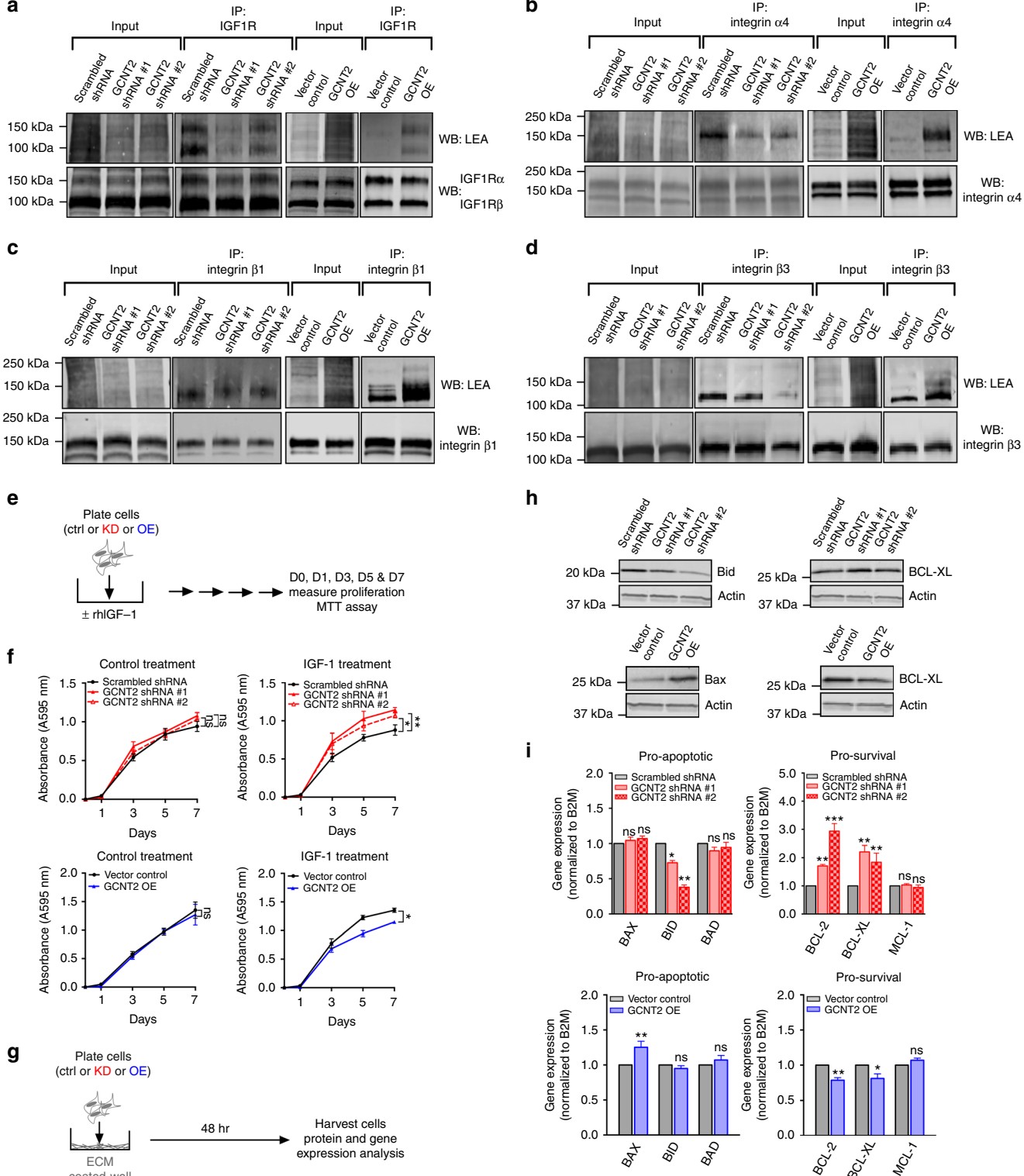

**Fig. 5** Growth factor and adhesion-mediated melanoma cell proliferation and survival are increased with loss of GCNT2. **a–d** Immunoblots of LEA lectin reactivity of IGF1R (**a**), integrin α4 (**b**), integrin β1 (**c**), and integrin β3 (**d**) subunits immunoprecipitated from UACC62 scrambled control and GCNT2 KD and A375 vector control and GCNT2 OE cell variants. **e** Schematic representation of IGF-1 growth assay. Cells were treated every 3 days with control or with 100 ng/ml recombinant human insulin-like growth factor-1 (rhIGF-1). Cell proliferation was measured using the MTT assay. **f** Cell proliferation measured by the MTT assay in UACC62 scrambled control and GCNT2 KD (top) cell variants and in A375 vector control and GCNT2 OE (bottom) cell variants. **g** Schematic representation of ECM-mediated cell survival assay. Cells were plated on an ECM model composed of fibronectin, laminin, and collagen I for 48 h. Cells were lysed for protein or gene expression analysis by western blot or qRT-PCR. **h, i** Cell survival family protein (**h**) and gene (**i**) expression measured by western blot and qRT-PCR in UACC62 scrambled control and GCNT2 KD (top) cell variants and in A375 vector control and GCNT2 OE (bottom) cell variants. Statistical analyses were done using two-way ANOVA with Dunnett's or Bonferroni's multiple comparison test or unpaired two-tailed Student's $t$ test. Results are representative of $n = 3$ experiments (mean ± SEM; $**p < 0.01$; $***p < 0.001$). See also Supplementary Fig. 5

GCNT2/I-branched glycan modifications on IGF1R could help elicit melanoma growth inhibition by decreasing IGF1R signaling activity.

To dissect how integrin:ECM engagement regulates melanoma cell survival, we plated GCNT2 KD, GCNT2 OE, and control melanoma cells on ECM and analyzed integrin:ECM induced signaling (Fig. 6d). KD of GCNT2 and loss of I-branched glycans enhanced overall cell tyrosine phosphorylation, as well as focal adhesion kinase (FAK) tyrosine phosphorylation and downstream target extracellular signal-regulated kinases (ERK1/2)

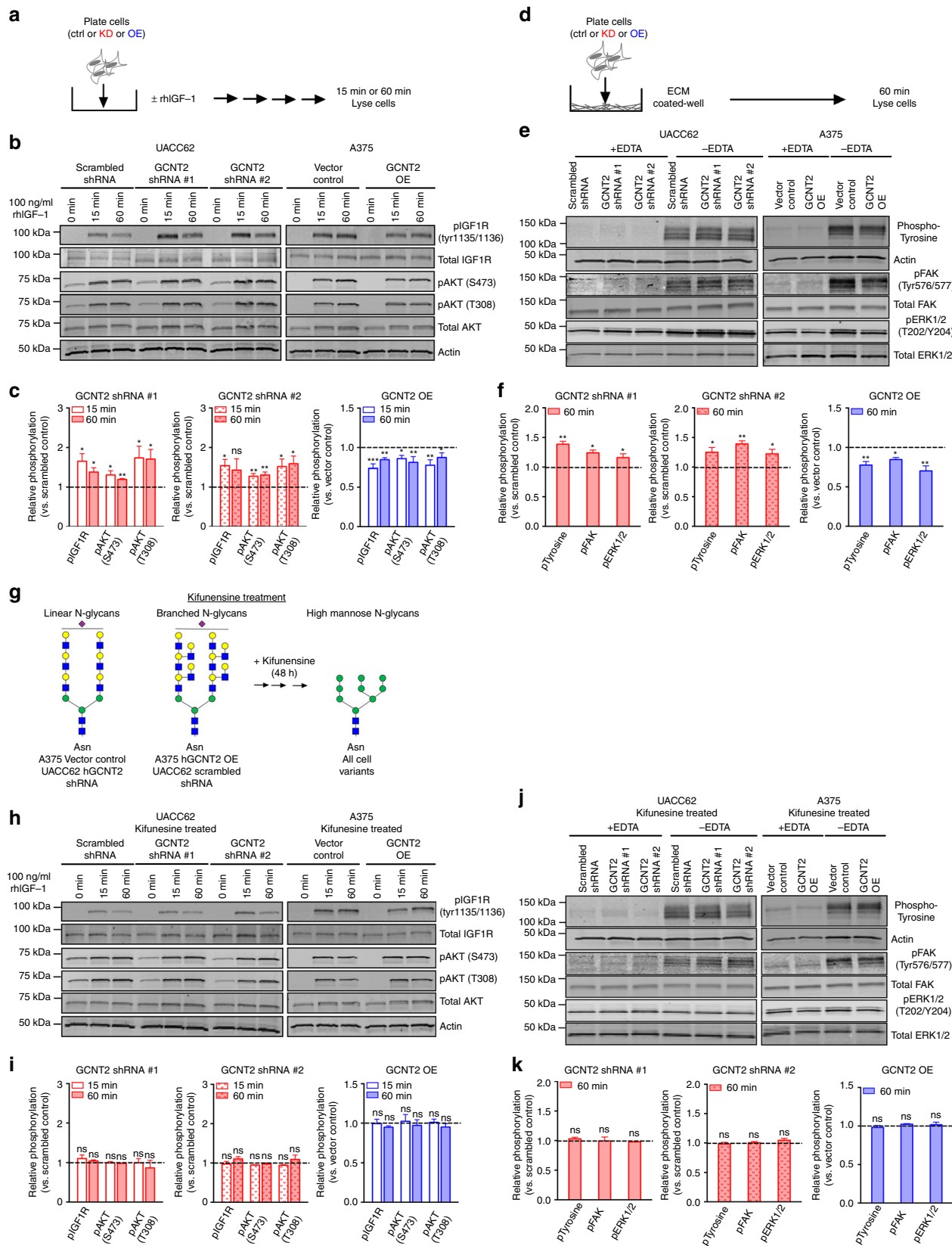

threonine and tyrosine phosphorylation (Fig. 6e, f), all of which are known to regulate cell survival[35,41]. Conversely, overexpression of GCNT2/I-branched glycans decreased overall cell tyrosine phosphorylation as well as FAK tyrosine phosphorylation, and downstream ERK1/2 threonine and tyrosine phosphorylation (Fig. 6e, f). Additionally, ethylenediaminetetraacetic acid (EDTA), a chelator of divalent cations necessary for integrin activation, completely blocked tyrosine phosphorylation, supporting an integrin-dependent signaling activity (Fig. 6e). Moreover, there was no consistent significant difference in either gene or cell-surface protein expression of commonly expressed integrin α and β chains in GCNT2 KD and GCNT2 OE cell variants compared to control cells (Supplementary Fig. 6e–p), suggesting that cell-surface expression differences between GCNT2 KD and GCNT2 OE cells and their controls cannot account for the observed altered signaling. Taken together, these results suggest that GCNT2/I-branching may inhibit melanoma survival, in part, by decreasing integrin-mediated cell signaling.

To reaffirm that differential GCNT2/I-branched glycan expression regulates IGF1R and integrin signaling, we treated GCNT2 KD, GCNT2 OE, and control cell variants with kifunensine, an inhibitor of α-mannosidase, a key enzyme required for complex N-glycan formation (Fig. 6g). Kifunensine treatment of GCNT2 KD, GCNT2 OE and control cell variants was used to prevent the formation of I-branched and i-linear glycans and normalize N-glycan structures between cell lines. Inhibition was confirmed by flow cytometry staining with the lectin PHA-L, which favors binding to complex N-glycans (Supplementary Fig. 6q, r). Critically, kifunensine treatment and N-glycan normalization ablated differences in IGF1R tyrosine phosphorylation and AKT serine and threonine phosphorylation (Fig. 6h, i), as well as ECM-dependent cell tyrosine phosphorylation, FAK tyrosine phosphorylation and ERK1/2 phosphorylation (Fig. 6j, k). Collectively, these data indicate that the modification of complex N-glycans with I-branched or i-linear glycan moieties elicits differential IGF1R and integrin signaling in melanoma cells.

**GCNT2 alters IGF-1- and RGD-binding activity on melanoma cells.** Next, we wanted to address how differential GCNT2/I-branched glycan expression could regulate IGF1R- and integrin-mediated signaling. Though glycans have primarily been shown to alter the expression of cell-surface receptors to modulate cellular responses[17,22,42], we did not observe consistent statistically significant differences in cell-surface expression of IGF1R or various integrin α and β receptors in GCNT2 KD, GCNT2 OE, or control melanoma cells (Supplementary Fig. 6a–p). Of note, cell-surface expression of integrin β3 in one of the two GCNT2 KD melanoma cell variants, as well as cell-surface expression of integrin αV in both GCNT2 KD melanoma cell variants was significantly decreased compared to control cells. Nonetheless, we do not believe that these differences are due to GCNT2/I-branched or i-linear glycan expression, as we still observed these alterations in cell-surface expression of integrin β3 and integrin αV in kifunensine treated GCNT2 KD and control cells (Supplementary Fig. 6s, t). We also did not observe these differences in expression in GCNT2 OE and control cells.

PolyLacNAc glycans on cell-surface receptors often serve as ligands for β-galactoside binding lectins, known as galectins. In cancer, galectin-1 (Gal-1) and galectin-3 (Gal-3) are the most well-studied galectins and both have been shown to enhance cell proliferation, signaling, and migration[43–45]. Moreover, melanoma cells have high expression of both Gal-1 and Gal-3 compared to other malignancies (Supplementary Fig. 7a). Because GCNT2 modifies polyLacNAcs, we hypothesized that GCNT2/I-branched glycans may control Gal-1- and/or Gal-3-binding in melanoma cells, thereby regulating downstream IGF1R and integrin-mediated signaling pathways. While we did not observe a significant decrease in Gal-1-binding (Supplementary Fig. 7b, c), we did observe a significant decrease in both exogenous and endogenous Gal-3-binding to GCNT2/I-branched glycan expressing melanoma cells (Supplementary Fig. 7d–g). To examine if differences in Gal-3-binding may be responsible for regulating differential downstream IGF1R and integrin-mediated signaling, we treated GCNT2 KD, GCNT2 OE and control cell variants with lactose, a competitive inhibitor of Gal-3-binding, or a control sugar, sucrose. Although lactose treatment normalized endogenous Gal-3-binding differences (Supplementary Fig. 7f, g), we still observed differential IGF1R and FAK phosphorylation in GCNT2 KD and GCNT2 OE cells compared to control cells (Supplementary Fig. 7h–k). Additionally, exogenous Gal-3 did not affect IGF1R or integrin-mediated signaling in GCNT2 KD or GCNT2 OE cell variants. Thus, while GCNT2/I-branched glycans decrease Gal-3-binding, we do not believe that this inhibition is responsible for mediating the observed reduction in IGF1R and integrin signaling in melanoma cells.

Intriguingly, the IGF-1 binding site of IGF1R, which has been mapped to residues 223–274 of human IGF1Rα[46], contains one N-linked glycan site. Additionally, though the ligand binding sites for integrins are not as well defined, there are multiple N-linked glycan sites located in the β-propeller repeat region of integrin α chains and in the βA domain of integrin β chains, where ligands are likely to bind[37]. Thus, we hypothesized that perhaps the addition of I-branches to N-linked glycans on the receptors may regulate ligand binding activity. To test this notion, we incubated control and GCNT2 OE cells with near-infrared labeled IGF-1 or RGD (arginine–glycine–aspartic acid motif in fibronectin) and

**Fig. 6** Loss of GCNT2 enhances IGF-1-induced PI3K and ECM-induced MAPK signaling pathways. **a** Schematic of IGF1R:IGF-1-induced melanoma signaling. Cells were treated with 100 ng/ml rhIGF-1 and lysed in RIPA buffer. **b, c** Immunoblot (**b**) and quantitation (**c**) of phosphorylated and total IGF1R and AKT in UACC62 scrambled control and GCNT2 KD (left) cell variants and A375 vector control and GCNT2 OE (right) cell variants treated with rhIGF-1 for 15 or 60 min. Quantitation of phosphorylated proteins was normalized to total protein expression and then compared to control cells. **d** Schematic of integrin: ECM-induced melanoma signaling. Cells were plated on an ECM mixture composed of fibronectin, laminin, and collagen I, and lysed in RIPA buffer. **e, f** Immunoblot (**e**) and quantitation (**f**) of tyrosine phosphorylation and total actin, phosphorylated and total focal adhesion kinase (FAK) and phosphorylated and total ERK1/2 in UACC62 scrambled control and GCNT2 KD (left) cell variants and A375 vector control and GCNT2 OE (right) cell variants plated on ECM for 60 min. **g** Schematic representation of kifunensine treatment. **h, i** Immunoblot (**h**) and quantitation (**i**) of phosphorylated and total IGF1R and AKT in kifunensine treated UACC62 scrambled control and GCNT2 KD (left) cell variants and A375 vector control and GCNT2 OE (right) cell variants treated with rhIGF-1 for 15 or 60 min. **j, k** Immunoblot (**j**) and quantitation (**k**) of cell tyrosine phosphorylation and total actin, phosphorylated and total focal adhesion kinase (FAK) and phosphorylated and total ERK1/2 in kifunensine treated UACC62 scrambled control and GCNT2 KD (left) cell variants and A375 vector control and GCNT2 OE (right) cell variants plated on ECM for 60 min. Quantitation of phosphorylated proteins was normalized to total protein expression and then compared to control cells. Statistical analyses were done using one-way ANOVA with Dunnett's multiple comparisons test. Results are representative of $n = 3$ experiments (mean ± SEM; $^{**}p < 0.01$; $^{***}p < 0.001$). See also Supplementary Fig. 6

detected IGF-1- and RGD-binding by infrared imaging. When normalized to total protein control, we found that GCNT2 OE cells labeled significantly less (30–40%) with IGF-1 and RGD than on control cells (Fig. 7a–d). Importantly, kifunensine treatment of control and GCNT2 OE cells to normalize complex N-glycan structures showed no significant difference in cell labeling (Fig. 7e–h). Together, these data suggest that GCNT2/I-branch expression modulates IGF-1 and RGD binding activity on melanoma cells to potentially regulate IGF1R- and integrin-related signaling.

**Loss of GCNT2 increases in vivo signaling and survival proteins.** To strengthen the role of differential GCNT2/I-branched glycan expression in modulating tumor growth through cell signaling and survival pathways, we analyzed phosphorylation of AKT and gene and protein expression of several prosurvival and proapoptotic molecules from in vivo tumor xenografts harvested from NSG mice. Consistent with our in vitro findings, tumors with low expression of GCNT2/I-branched glycans displayed an increase in both serine and threonine AKT phosphorylation (Supplementary Fig. 7a–d). Additionally, we observed a decrease in proapoptotic Bid and Bax protein and gene expression from in vivo tumors with low-GCNT2/I-branched glycan expression (Supplementary Fig. 8e–j), as well as an increase in prosurvival BCL-XL protein and gene expression from in vivo tumors with low GCNT2/I-branched glycan expression (Supplementary Fig. 8k–p). Together, these data further support the role of GCNT2/I-branched glycans in regulating melanoma cell growth and survival.

In summary, these findings support a model demonstrating that, throughout malignant progression, melanomas decrease expression of GCNT2/I-branched glycans (Fig. 8a). Loss of GCNT2/I-branches increased melanoma xenograft growth and enhanced three-dimensional colony formation and survival. Further dissecting candidate N-glycosylated protein targets, IGF1R and integrins, revealed that GCNT2/I-branches inhibited IGF-1 and ECM-dependent proliferation, survival, and signaling (Fig. 8b). While effects associated with these N-glycoprotein targets act as surrogates for global N-glycan modifications, overall they implicate GCNT2/I-branched glycans as a negative regulator of melanoma growth (Fig. 8b).

## Discussion

Here, we report that, compared to NHEMs, melanomas downregulate the glycosyltransferase, GCNT2, and display a corresponding loss of I-branched glycans on the cell surface. Additionally, in clinical specimens, we found that GCNT2 expression inversely correlated with progression. Functionally, loss of GCNT2/I-branched glycans promoted melanoma xenograft growth, colony formation, and cell survival, while overexpression of GCNT2/I-branched glycans negatively regulated melanoma xenograft growth, colony formation, and cell survival. Additional studies focusing on a subset of potential N-glycan protein targets demonstrated that low GCNT2/I-branched glycan expression increased melanoma cell proliferation and survival by enhancing IGF1R- and integrin:ECM-mediated signaling. Moreover, we found that GCNT2/I-branched glycans decreased IGF-1 and RGD ligand binding activity on melanoma cells, suggesting that GCNT2/I-branches may modulate IGF1R and fibronectin: integrin signaling through modulation of ligand binding capacity. Overall, our study reveals that loss of GCNT2/I-branched glycans in melanomas regulates multiple cell surface glycoprotein signaling pathways and promotes melanoma growth and survival.

Our studies predominately focused on the presence or absence of I-branched glycans in regulating melanoma. Initial analysis of our MS results (Supplementary Fig. 1), showed that melanomas express longer polyLacNAcs than normal melanocytes. Thus, it is possible that the synthesis of longer polyLacNAcs is the more important contributor to malignancy. However, we would have expected to observe an increase in the expression of the β1,3-glucosaminyltransferase (B3GNT) genes, which encode enzymes that extend polyLacNAcs[47], in melanomas compared with normal melanocytes. Results of our glycomic gene expression array did not reveal an increase in any of the B3GNT genes. In fact, we actually found that normal melanocytes have higher expression of B3GNT9 compared to A375 and G361 melanoma cell lines, suggesting that NHEMs could also have longer poly-LacNAc glycans. Furthermore, data mining of three independent clinical cohorts[25–27] revealed no significant difference in expression of B3GNT1, B3GNT2, and B3GNT8 genes between normal melanocytes and melanomas. Because we did not observe a significant difference in B3GNT gene expression, we hypothesize that extended polyLacNAcs could potentially be regulated by B3GNT and GCNT2 competition for the same nucleotide donor sugar, UDP-GlcNAc, thereby limiting polyLacNAc length when GCNT2 is expressed. Still, in depth analysis of our MS data demonstrated that normal melanocytes can have extended polyLacNAcs, consisting of four or more LacNAc residues, which are likely to be modified with I-branched glycans (Supplementary Fig 1a–c—peaks 4939, 5388, and 5837). In our model, we believe that the downregulation of GCNT2 is important for the loss of I-branched glycans in melanoma cells and perhaps even for the increase in extended i-linear polyLacNAcs through less donor sugar competition between B3GNT and GCNT2.

Our data reveal that changes in glycome-related genes significantly correlate with melanoma progression. These results are supported by the fact that numerous studies have highlighted the importance of aberrant protein glycosylation in malignancy[1,7,8]. Increases in global sialylation[12], truncation of O-linked glycans[9], and increase in the size and complexity of N-linked glycans[15] have all been shown to regulate cancer cell signaling, growth and metastases, suggesting that changes in expression of glycan-forming and glycan-degrading genes can shape malignancy. While our data show that the I-branch glycan-forming enzyme, GCNT2 is downregulated in melanomas, previous studies in breast[48] and prostate[49] cancer have shown that GCNT2 expression is upregulated with progression and correlates with metastasis. These observed differences in GCNT2 expression during malignant progression, suggest that cancers from different cell lineages transcriptionally regulate their glycosylation-related gene expression differently to uniquely control their malignant phenotype. As cancer cells arise from a variety of cell types including epithelial, glial, melanocytic, stromal, and hematopoietic origins, our results highlight the importance of identifying specific glycomic gene signatures as mediators of a putative cancer phenotype.

A key finding in our studies is that melanomas significantly downregulate GCNT2 and I-branched glycans compared to NHEMs. An intriguing possibility for this disparity is that melanomas acquire these changes as part of a broader reversion to a more embryonic-like phenotype. In fact, erythrocytes, epithelial cells and dividing cells of the fetus predominantly express i-linear glycans, which are thought to promote cell adhesion and proliferation during development, whereas in adults, i-linear glycans on these cells are largely replaced by I-branched glycans[50–52]. Hence, the decrease in GCNT2 and increase in i-linear glycan expression on melanoma cells reflect a more dedifferentiated state compared to their normal counterpart. This is further supported by an accumulating body of evidence that cancer cells often possess glycomic and developmental traits reminiscent of those of normal stem cells. The hallmark traits of stem cells, including

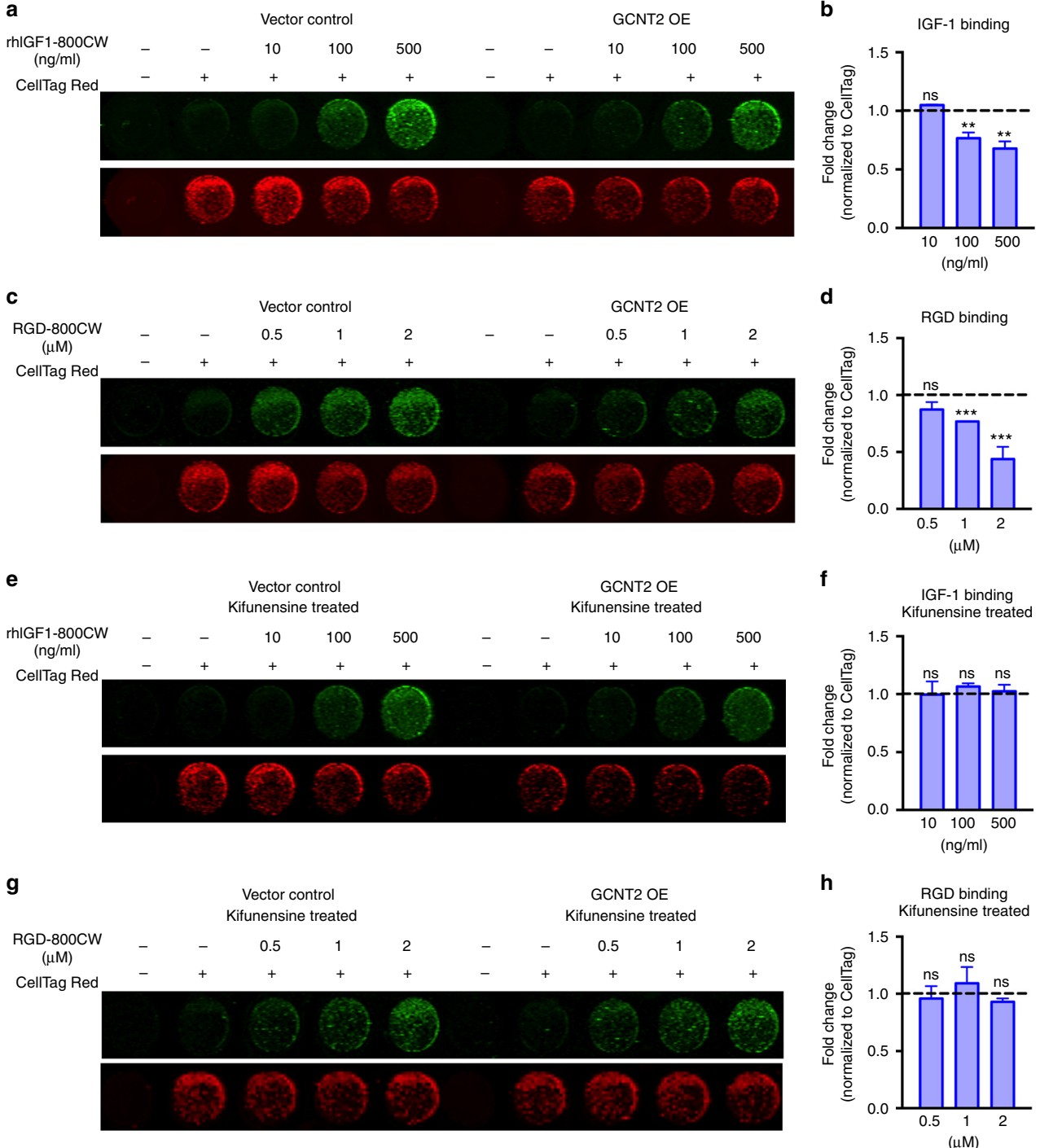

**Fig. 7** GCNT2 decreases IGF-1 and RGD-peptide ligand binding activity. **a**, **b** Representative images (**a**) and quantitation (**b**) of rhIGF1-800CW binding activity to A375 vector control and GCNT2 OE cell variants. **c**, **d** Representative images (**c**) and quantitation (**d**) of RGD-800CW binding activity to A375 vector control and GCNT2 OE cell variants. **e**, **f** Representative images (**e**) and quantitation (**f**) of rhIGF1-800CW binding activity to kifunensine treated A375 vector control and GCNT2 OE cell variants. **g**, **h** Representative images (**g**) and quantitation (**h**) of RGD-800CW binding activity to kifunensine-treated A375 vector control and GCNT2 OE cell variants. Quantitation of rhIGF1-800CW and RGD-800CW binding activity was normalized to total protein expression using CellTag700 Red Dye and then compared to control cells denoted by dashed line. Statistical analyses were done using unpaired two-tailed Student's $t$ test. Results are representative of $n = 3$ experiments (mean ± SEM; $^{**}p < 0.01$; $^{***}p < 0.001$)

self-renewal and differentiation capability, are paralleled by the high-proliferative capacity and heterogeneity of tumor cells[53]. Currently, a multitude of studies are focused on identifying the most virulent, stem-like cancer cells and their progeny within tumors, and perhaps loss of GCNT2 and a corresponding increase in the presence of i-linear glycans is an indication of a dedifferentiated melanoma cell.

The negative association of GCNT2 with metastasis suggests that loss of GCNT2/I-branched glycans may help melanomas progress. The majority of cancer deaths are attributed to the

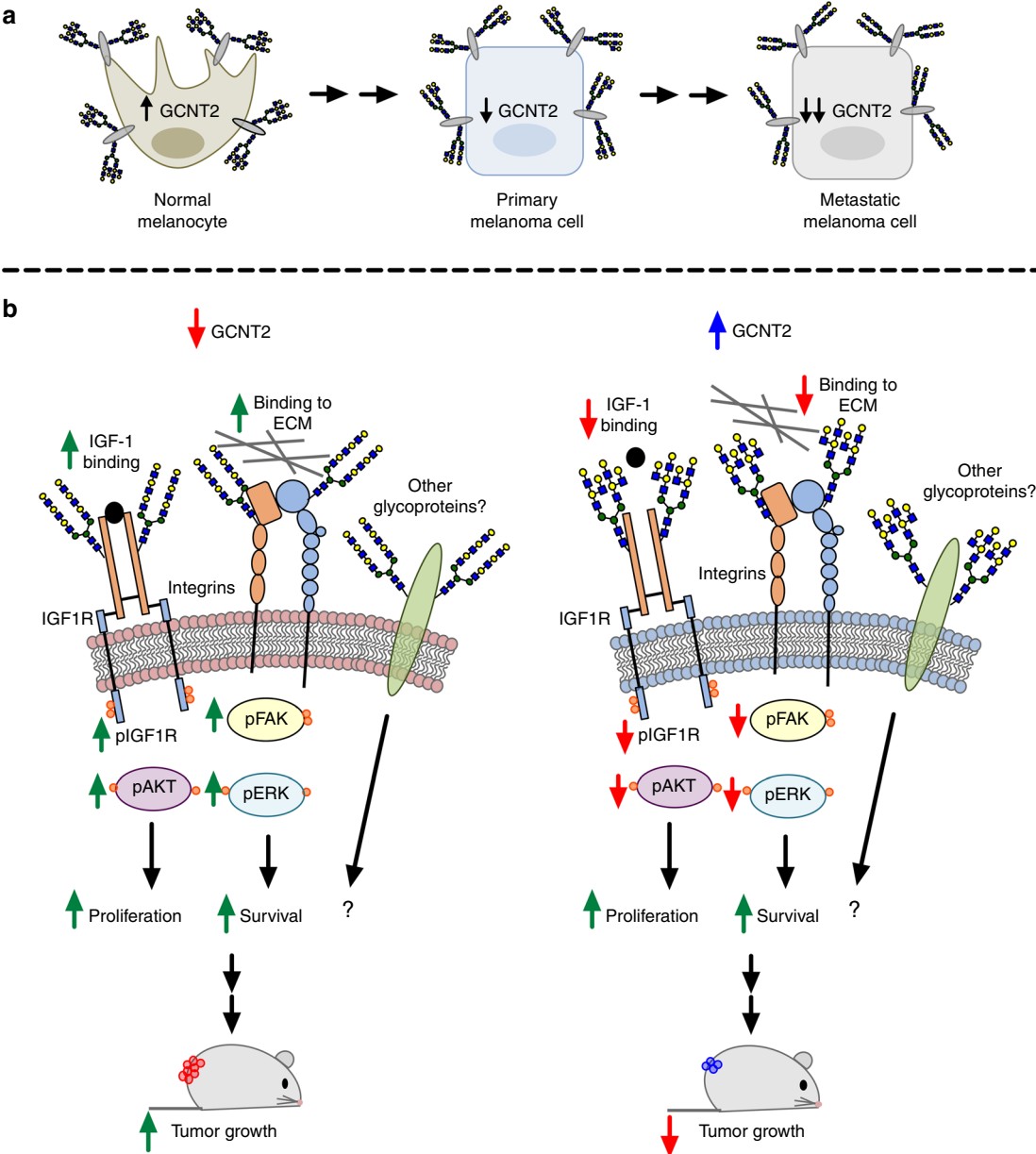

**Fig. 8** GCNT2 negatively regulates melanoma cell growth and survival. **a** Melanomas downregulate the I-branching glycosyltransferase, GCNT2, upon malignant transformation, leading to the formation of extended poly-N-acetyllactosamine (i-linear) glycans on cell-surface glycoproteins. **b** Loss of GCNT2/I-branched glycans promotes melanoma growth and survival and enhances growth factor receptor and integrin-mediated cell proliferation and survival signaling pathways

metastatic spread of cancer cells to visceral organs rather than to the primary tumor growth[54]. Increasing evidence suggests that tumor cells remodel their cell-surface glycans to aid in the metastatic process by promoting dynamic interactions with ECM, migration through the circulation and lodgment/entry into distant tissues[1,11,13,14]. Our data further implicate tumor cell i-linear/I-branched glycan modifications as regulators of metastatic potential by controlling prosurvival and cell signaling activities. As such, detection of GCNT2/I-branched glycans provides intriguing insights into its potential use as a biomarker to help predict which patients are at risk for progression to metastatic disease.

While we have demonstrated that I-branched glycans negatively regulate IGF1R- and integrin-mediated cell growth and survival signaling pathways in melanoma, the opposite outcome is also possible for the function of other glycoproteins. It has been

shown that the same glycan modification may have a different function depending on which protein is modified. For example, addition of a bisecting GlcNAc on N-glycans by Mgat3 to EGFR increased EGFR signaling and reduced cell adhesion, promoting metastasis[55]. However, addition of a bisecting GlcNAc to N-glycans on E-cadherin increased cells adhesion and inhibited migration[56]. Since glycosyltransferases are able to modify many proteins, it is likely that GCNT2 modifies additional glycoproteins. Thus, it is possible that presence of I-branched glycans on a different glycoprotein may actually enhance, rather than inhibit downstream signaling pathways of that protein. Therefore, a more complete delineation of GCNT2 glycoprotein targets in melanoma cells is needed, as this should help to further define the role of the glycome in malignancy.

Because polyLacNAcs often serve as ligands for β-galactoside-binding lectins, known as galectins, and GCNT2 modifies

polyLacNAcs, we explored whether promelanoma galectin (Gal)-1 and Gal-3 regulate differential IGF1R and integrin-ECM signaling. Interestingly, we observed significant reduction in Gal-3 but not Gal-1 binding (Supplementary Fig. 7b–g). However, to our surprise, we did not observe any Gal-3-dependent IGF1R- or integrin-ECM-mediated signaling differences in melanoma cells expressing i-linear or I-branched polyLacNAc glycans (Supplementary Fig. 7h–k). While this finding is paradoxical to the role of galectins in helping to mediate cellular signaling events, we believe differential Gal-3-binding is playing a significant role in other areas of melanoma malignancy. This also argues for the hypothesis that specific glycoprotein pathways in various cancer subtypes are differentially controlled by their N-glycan structure and ability to bind galectins. Thus, it is likely the "net" cellular response that ultimately governs virulent behavior.

Though we found that GCNT2/I-branched glycan expression decreased IGF-1 and RGD ligand binding activities on the surface of melanoma cells, there are several potential reasons for how GCNT2/I-branched glycans regulate downstream signaling, including but not limited to, cell surface receptor expression level, receptor-ligand binding, and membrane organization, including dimerization and clustering. Though we do not consistently observe cell-surface expression differences in direct relation with suppressed or elevated signaling, one possibility is that GCNT2/I-branched glycans displayed by N-glycans at key glycosylation sites on particular glycoproteins, such as IGF1R or integrins, may influence ligand induced conformational changes required for receptor signal transduction. Alternatively, as many cell surface receptors, including IGF1R and integrins, undergo receptor dimerization or clustering upon ligand binding, presence of I-branched glycans, could inhibit the ability of these receptors to complex efficiently.

Lastly, further understanding of how GCNT2 and I-branched glycans regulate growth factor receptor and integrin receptor signaling may provide important insights into the design of novel cancer therapies that either directly target glycans or predict efficacy of certain therapies based on receptor glycosylation. Also, because most cell surface proteins are glycosylated, targeting cancer-associated glycans may prove to be a valuable treatment option. Currently, multiple strategies to target glycans for cancer therapy are being developed. Vaccines against several cancer-associated glycans are in development with the hope of eliciting an immune response specifically to malignant cells[57]. In addition, targeting the cancer-associated Tn glycoform of MUC1 using genetically modified T cells expressing a chimeric receptor (CAR T cells) has shown efficacy in xenograft models of T cell leukemia and pancreatic cancer[58]. Our results further support the idea that targeting cancer-specific glycans rather than individual glycoproteins may be a more effective treatment strategy.

In conclusion, we have established that melanomas harness distinct glycome genes to transcriptionally regulate glycosylation-associated malignancy. Furthermore, these data demonstrate that I-branch modifications to N-glycans by GCNT2 can lead to a global change in cell surface glycosylation that significantly alters melanoma cell growth, survival, and various signaling pathways. These findings highlight the importance of studying cancer-associated glycome to identify drivers of malignancy that can be targeted in novel cancer therapeutics or utilized as biomarkers of malignant progression in patients with latent disease.

## Methods

**Contact for reagent and resource sharing**. Further information and requests for reagents may be directed to, and will be fulfilled by the corresponding author Charles J. Dimitroff, Ph.D. (cdimitroff@bwh.harvard.edu).

**Experimental model and subject details**. Cell lines: Human melanoma A375, G361, UACC257, UACC62, SKMEL28, SKMEL2, and MeWo cell lines were maintained in Dulbecco's Modified Eagle Medium (DMEM) supplemented with 10% fetal bovine serum (FBS), 1% penicillin/streptomycin and 1% L-glutamine (Gibco) and either purchased from ATCC or validated for expression of melanocytic/pigmentation differentiation markers. We did not use any misidentified cell lines as indicated by NCBI Biosample. All lines were tested for mycoplasma and found to be negative in our laboratory. NHEM were obtained from Promo-Cell (#C-14043, Heidelberg, Germany) and cultured ≤2 weeks in Melanocyte Growth Medium M2 (#C-12453, Promo-Cell). All cells were maintained at 37 °C with 5% $CO_2$.

To generate GCNT2 KD variants lentiviral shRNA constructs in the pLKO.1puro vector (#10878, Addgene) against GCNT2 (NM_145649) or scrambled controls were purchased from the Mission collection (Sigma-Aldrich). Lentiviral supernatants were generated by co-transfection of helper plasmids, pN8e-GagPolΔ8.1 and pNE8e-VSV/G, in the packaging cell line, HEK293-EBNA. Viral supernatants were harvested 48–72 h after transfection and UACC62 melanoma cells were transduced in the presence of 6 µg/ml polybrene (#TR-1003-G, EMD Millipore) and selected with 1 µg/ml puromycin (#61-385-RA, Corning) to generate stable KD cell lines. KD of human GCNT2 was achieved with the shRNA target sequences—shRNA #1: 5′-GCTAACAAGTTTGAGCTTAAT-3′ and shRNA #2: 5′-GCTCACCTCTATATTAGTTTA-3′.

To generate GCNT2 OE cell variants full length human GCNT2 cDNA (kindly provided by Tong Hao, Dana Farber Cancer Institute) was amplified using Platinum PCR SuperMix High Fidelity (#12532016, Life Technologies), digested and ligated into the retroviral plasmid, pLNCX2 (#631503, Clontech), using the Rapid DNA Ligation Kit (Roche) according to manufacturer's protocol. Sequences were validated using the human GCNT2 cloning primers (Supplementary Table 1). Empty vectors were used as controls. Retroviral vectors and two helper plasmids, pN8e-GagPolΔS and pN8e-VSV/G, were transfected into the HEK293-EBNA packaging cell line. Viral supernatants were harvested 48–72 h after transfection and human A375 melanoma cells were infected with filtered retroviral supernatant and selected in 500 µg/ml Geneticin (#11811023, ThermoFisher).

**Mice**. Nonobese diabetic/severe combined immunodeficiency interleukin (IL)-2Rγ$^{−/−}$ (NSG) mice were purchased from The Jackson Laboratories (#005557, Bar Harbor, ME). Age and sex-matched mice that were at least 6 weeks of age were used for all experiments. All mice were used according to the Brigham and Women's Hospital Center for Comparative Medicine, National Institutes of Animal Healthcare Guidelines, and Institutional Animal Care and Use Committee approval No. 2016N000086.

For xenografts, human melanoma A375 vector control and GCNT2 OE cell variants and human melanoma UACC62 scrambled shRNA and GCNT2 KD (both shRNA #1 or #2) cell variants were injected subcutaneously ($1 \times 10^6$ cells/inoculum) into the flanks of recipient NSG mice. Tumor growth was assessed with calipers every 2–4 days and tumor volume was calculated using the formula [tumor volume (mm$^3$) = π/6 × 0.5 × length × (width)$^2$][59]. All in vivo experiments were monitored in a nonblinded fashion, no method of randomization was used and no mice were excluded in any experiment. Tumors were harvested 24 days after tumor cell inoculation, weighed, and taken for protein lysates or RNA purification. Experiments were repeated >3 times and data were taken from ≥20 tumors/cell type.

**Glycome gene set enrichment analysis**. Publicly available replicate Affymetrix HuExon1.0ST gene expression data from the NCI-60 cancer cell line collection (GSE29682) were extracted on the gene-level using expression console and the robust multi-array (RMA)-method. Gene-specific signals were averaged among the replicates for each cell line. Gene-set enrichment analysis was performed using GSEA v2 using a signature of the annotated human glycome-related genes (698 genes—glycosyltransferases, glycosidases, glycoproteins, proteoglycans, nucleotide-sugar transporters, nucleotide-sugar forming enzymes, sulfotransferases, and lectins) contained in The GlycoV4 oligonucleotide array (Affymetrix) from the NIH Consortium for Functional Glycomics.

**Glycome gene expression profiling**. RNA was isolated from melanocyte/melanoma cell pellets using the RNAEasy Mini Kit (#74104, Qiagen) per manufacturers protocol. Gene expression was analyzed on a GlycoV4 oligonucleotide array. A full description of the GlycoV4 array is available at http://www.functionalglycomics.org/static/consortium/resources/resourcecoree.shtml.

The raw microarray data were processed using R package affy[60] and the RMA[61] method was used to normalize the data and calculate the expression values. Statistical analysis were done with limma[62], and Benjamini–Hochberg correction[63] was used to adjust the p values. Only genes with absolute fold change >2 and adjusted p value < 0.05 were considered differentially expressed. For heat map generation, the log$^2$ transformed expression values of each gene were standardized to mean 0 and standard deviation of 1. Data were deposited into GEO (GSE94837) and is publicly viewable at https://www.ncbi.nlm.nih.gov/geo/query/acc.cgi?token=oxcpoysofhizvst&acc=GSE94837.

**Glycomic profiling**. For N-glycan structural analysis, all cells were treated as described previously[64,65]. Briefly, cell pellets ($2 \times 10^6$ for NHEM and $25 \times 10^6$ for A375/G361) were subjected to sonication in the presence of detergent (CHAPS), reduced in 4 M guanidine-HCl (Pierce), carboxymethylated, and digested with trypsin. The digested glycoproteins were then purified by C18-Sep-Pak (Waters Corp., Hertfordshire, UK). N-glycans were released by peptide N-glycosidase F (E. C. 3.5.1.52; Roche Applied Science) digestion. Released N-glycans were permethylated using the sodium hydroxide procedure and purified by C18-Sep-Pak. Where indicated, cells were first digested with sialidase-S (*Streptococcus pneumoniae*; E.C. 3.2.1.18; Prozyme, GK80020) to cleave α2,3 sialic acids (NeuAc). Released N-glycans were incubated in 200 μl of 50 mM sodium acetate (37 °C, pH 5.5). One hundred and seventy milliunits of the enzyme were added to the sample for 24 h. The results shown are representative of three independent cell glycan preparations.

To analyze the structure of released glycans, matrix-assisted laser desorption ionization-time of flight MS (MALDI-TOF MS) and MALDI-TOF/TOF MS/MS were performed. MS data were acquired using either a Voyager-DE STR MALDI-TOF or a 4800 MALDI-TOF/TOF (Applied Biosystems) mass spectrometer. MS/MS data were acquired using a 4800 MALDI-TOF/TOF mass spectrometer. Permethylated samples were dissolved in 10 μl of methanol, and 1 μl of dissolved sample was premixed with 1 μl of matrix (Voyager, 20 mg/ml 2,5-dihydroxybenzoic acid in 70% (v/v) aqueous methanol; TOF–TOF, 20 mg/ml 3,4-diaminobenzophenone in 75% (v/v) aqueous MeCN, spotted onto a target plate, and dried under vacuum. For the MS/MS studies, the collision energy was set to 1 kV, and argon was used as collision gas. The 4700 Calibration standard kit, calmix (Applied Biosystems), was used as the external calibrant for the MS mode, and [Glu1] fibrinopeptide B human (Sigma) was used as an external calibrant for the MS/MS mode.

Analyses of MALDI Data—the MS and MS/MS data were processed using Data Explorer 4.9 Software (Applied Biosystems). The processed spectra were subjected to manual assignment and annotation with the aid of a glycobioinformatics tool, GlycoWorkBench[66]. The proposed assignments for the selected peaks were based on $^{12}$C isotopic composition together with knowledge of the biosynthetic pathways. The proposed structures were then confirmed by data obtained from MS/MS and linkage analysis experiments.

Further glycan structure interrogation was performed using gas chromatography–MS (GC–MS) linkage analysis. Partially methylated alditol acetates were prepared as previously described[64]. Linkage analysis of partially methylated alditol acetates was performed on a PerkinElmer Life Sciences Clarus 500 instrument fitted with a RTX-5 fused silica capillary column (30 mm × 0.32 mm inner diameter; Restek Corp.). The sample was dissolved in 20–50 μl of hexanes and injected manually (2–3 μl). Injector temperature was set at 250 °C. Partially methylated alditol acetates were eluted with the following linear gradient oven. Initially the oven temperature was set at 65 °C for 1 min, heated to 290 °C at a rate of 8 °C per min, held at 290 °C for 5 min, and finally heated to 300 °C at a rate of 10 °C per min.

**RNA extraction and real time qRT-PCR**. Total RNA was isolated from cells using the RNAEasy Mini Kit (#74104, Qiagen) according to the manufacturers protocol. cDNA was made using SuperScript VILO cDNA synthesis kit (#11754250, ThermoFisher). Samples were assayed on a StepOne Plus Real-Time PCR System (Applied Biosystems) using Fast SYBR Green Master Mix (#4385612, ThermoFisher). The cycling conditions were 95 °C for 20 s, followed by 40 cycles at 95 °C for 10 s, 58 °C for 30 s and the relative amounts of transcripts were analyzed using the $2^{(-\Delta\Delta Ct)}$ method[67]. Primers used for real time qPCR are cataloged in Supplementary Table 1.

**Flow cytometry**. Antibodies and concentrations used for flow cytometry are cataloged in Supplementary Table 2. Analysis of cell surface I-branched and i-linear glycans on NHEMs and human melanoma cell lines was performed using monoclonal antibodies, human IgM OSK-28 i-, and OSK-14 I-antibodies. The cells were first treated with 125 mU/ml neuraminidase (#10269611001, Roche) in PBS for 1 h at room temperature. The cells were placed for 1 h on ice and then subsequently stained with secondary antibody, antihuman IgM conjugated to APC (Biolegend) for 30 min on ice followed by live/dead cell marker 7AAD (Biolegend) for 10 min at room temperature. The analysis of cell-surface glycans on GCNT2 KD and GCNT2 OE melanoma cell variants was performed using 0.5 μg/ml SNA, MAL-II, PHA-L, STA, and LEA lectins conjugated to biotin (Vector Labs). The cells stained for 1 h on ice and then subsequently stained with secondary antibody, Streptavidin conjugated to APC for 30 min on ice followed by live/dead cell marker 7AAD for 10 min at room temperature. Cell surface staining of IGF1R conjugated to PE and Integrins α4, α6, αV, β1, and β3 all conjugated to APC (Biolegend) was performed. The cells stained for 30 min on ice and then stained with live/dead cell marker Zombie or with 7AAD (Biolegend) for 10 min at room temperature. Cells were stained with recombinant human Galectin-1 and Galectin-3 (Peprotech) ± 100 mM Lactose (Sigma) for 60 min on ice. Cells were then stained with anti-human Galectin-1 (R&D) followed by anti-goat-APC or with anti-human Galectin-3 conjugated to Alexa647 (also used for endogenous Gal-3 detection on cell surface) for 30 min on ice. Finally, cells were stained with 7AAD (Biolegend) for 10 min at room temperature. All cells were analyzed on a FACSCanto (Becton Dickinson).

**Immunohistochemistry**. Antibodies, concentrations and reagents used for immunohistochemistry are cataloged in Supplementary Table 2. Sections of archival FFPE normal human skin ($n = 4$), benign melanocytic nevi ($n = 3$), primary cutaneous melanoma ($n = 4$), and metastatic melanoma ($n = 4$) specimens were obtained in accordance with institutional review board (IRB) approval. This research was deemed exempt as determined IRB review panel. Furthermore, sections from a human melanoma progression TMA were kindly provided by Dr. Lyn Duncan (Massachusetts General Hospital, Boston, MA)[28]. This TMA contained 480 0.6 mm cores of benign nevi ($n = 132$), primary thin (<1.0 mm) and thick (>2.0 mm) melanomas ($n = 198$), lymph node metastases ($n = 58$), and visceral metastases ($n = 92$). Sections were deparaffinized in xylene and subsequently rehydrated with 100%, 95%, and 75% ethanol and deionized water. Sections were then placed in antigen retrieval solution and boiled at 100 °C for 20 min. Sections were then stained with a 1:500 dilution of GCNT2 antibody (Sigma-Aldrich) for 30 min at 37 °C. GCNT2 primary antibody was detected using the Leica Bond Polymer Refine Detection Kit (Leica #DS9800), the polymer-horse radish peroxidase secondary antibody is incubated for 15 min at room temperature. For dual IHC staining, following GCNT2 staining (in brown), select sections were stained with MITF primary antibody (Leica) for 30 min at 37 °C. MITF primary antibody was detected (in red) using the Leica Bond Polymer Refine Red Detection Kit (Leica #DS9390). Detection is performed by incubating with post primary alkaline phosphatase (AP) for 15 min at room temperature, followed by polymer-AP for 20 min at room temperature. All sections were counterstained in hematoxylin. Images were acquired using a Nikon eclipse Ti microscope and a Nikon FDX-35 digital camera.

Dual GCNT2/MITF stained specimens were analyzed semiquantitatively as follows: random fields totally >50 MITF$^+$ melanocytes or >100 MITF$^+$ melanoma cells were scored as 0, 1 (1–25% cells positive for GCNT2), 2 (25–50% cells positive for GCNT2), 3 (50–75% cells positive for GCNT2), 4 (75–100% cells positive for GCNT2), and data were represented as % GCNT2$^+$/MITF$^+$. For TMA scoring, individual cores in GCNT2-stained TMA cores were first excluded if melanocytes/melanoma cells were absent or tissue quality deemed unsuitable by pathologist, melanocyte/melanoma cells (as identified/confirmed by a pathologist) were graded and scored as above. All IHC scoring was performed in a blinded manner by a technician, who did not know the tumor grade, stage or site of metastasis, under the guidance of a dermatopathologist.

**Glycosidase and metabolic inhibitors of glycosylation and glycolipid synthesis**. To cleave N-glycans, cells were lysed in NP-40 lysis buffer and treated with peptide-N-glycosidase F (PNGase F) as per manufacturer's protocol (#P0704S, New England Biolabs). To inhibit de novo complex N-glycan formation, melanoma cell cultures were treated with water control or 1 μg/ml mannosidase I inhibitor, kifunensine (#K1140, Sigma-Aldrich), for 48 h in DMEM/10%FBS/1%P/S at 37 °C. To inhibit glycolipid synthesis, cell cultures were treated with 2 μM of PPPP or a control PPPP isomer (Gift from Ronald L. Schnaar, Johns Hopkins University) for 48 h in DMEM/10%FBS/1%P/S at 37 °C.

**Methylcellulose assays**. Methylcellulose Stock Solution (#HSC001, R&D) was thawed overnight at 4 °C and prepared according to manufacturer's protocol. Cells were plated in triplicate at 100 cells/well in 1% methylcellulose in 24-well ultra-low attachment plates (#3473, Costar) in DMEM/10%FBS/1%P/S for 12 days. Cells were refed with 100 μl of DMEM/10%FBS/1%P/S every 3 days. On Day 12, cell colonies were counted and photographs taken at 4×. After photographs were taken, plates were moved to 4 °C for about 30 min. Then 1 ml of DMEM/10%FBS/1%P/S was added to each well. All cells were collected and put into a 15 ml conical tube, 10 ml of excess media was added, cells were centrifuged at 1200 RPM for 10 mins and washed 2× with DMEM/10%FBS/1%P/S. Washed cells were split for RNA extraction and flow cytometry analysis of dead cells using the Zombie stain from Biolegend (see above protocols).

**IGF1R:IGF-1 cell proliferation assays**. Cells were plated onto tissue culture grade six-well plates in DMEM containing 10% FBS for 24 h. Cells were then treated every 3 days with or without 100 ng/ml recombinant human IGF-1 (#100-11, Peprotech) in DMEM containing 1% FBS and placed at 37 °C. On Day 0 (baseline), 1, 3, 5, and 7 cell proliferation was measured using the 3-(4,5-dimethylthiazol-2-Yl)-2,5-diphenyltetrazolium bromide (MTT) assay (#4890-25-02, Trevigen) according to the manufacturers protocol.

**Adhesion survival assays**. Tissue culture grade 24-well plates were coated overnight with an ECM model mixture containing 0.5 μg/cm$^2$ laminin from human fibroblasts (#L4544, Sigma-Aldrich), 2.5 μg/cm$^2$ fibronectin from human plasma (#F2006, Sigma-Aldrich) and 6 μg/cm$^2$ collagen I from rat tail (#C7661, Sigma-Aldrich). Wells were blocked for 30 min with 1% bovine serum albumin to block nonspecific binding and washed 3× with serum-free DMEM prior to addition of cells. Cells were harvested using 1 mM EDTA, washed 2× with serum-free DMEM. Cells were added to ECM coated plates in serum-free DMEM and incubated for 48 h at 37 °C. At end of incubation, cells were lysed either in RIPA Buffer (#89900, ThermoFisher) in the presence of protease/phosphatase inhibitors (#88668, ThermoFisher) or in Buffer RLT (Qiagen). Cells were analyzed by western blot

according to protocol below or RNA was isolated for qRT-PCR analysis as described above.

**IGF1R signaling assays**. Cells were plated into tissue culture grade six-well plates and placed into serum-free DMEM for 24 h. Cells were then treated with 100 ng/ml recombinant human IGF-1 (#100-11, Peprotech) for 15 min or 60 min in serum-free DMEM at 37 °C. At end of incubation plates were placed on ice and processed for western blotting as described below.

**Galectin IGF1R signaling assays**. Cells were plated into tissue culture grade six-well plated and incubated for 24 h in DMEM/10%FBS/1%PenStrep media supplemented with no additional sugar, 50 mM sucrose or 50 mM lactose. Cells were placed in serum-free DMEM for at least 4 h and then treated with 100 ng/ml recombinant human IGF-1 (Peprotech) for 60 min in serum-free DMEM with no additional sugar, 10 mM sucrose or 10 mM lactose at 37 °C. At end of incubation plates were placed on ice and processed for western blotting as described below.

**Adhesion signaling assays**. Tissue culture grade 24-well plates were coated overnight with an ECM model mixture containing $0.5 \, \mu g/cm^2$ laminin from human fibroblasts (#L4544, Sigma-Aldrich), $2.5 \, \mu g/cm^2$ fibronectin from human plasma (#F2006, Sigma-Aldrich), and $6 \, \mu g/cm^2$ collagen I from rat tail (#C7661, Sigma-Aldrich). Wells were blocked for 30 min with 1% BSA to block nonspecific binding and washed 3× with serum-free DMEM prior to addition of cells. Cells were harvested using 1 mM EDTA, washed 2× with serum-free DMEM and incubated with or without 10 mM EDTA (#15575-020, ThermoFisher), to inhibit integrin activity, for 30 min in serum-free DMEM. Cells were added in the presence or absence of 10 mM EDTA to ECM coated plates and incubated for 60 min at 37 °C. At end of incubation plates were placed on ice and processed for western blotting as described below.

**Galectin adhesion signaling assays**. Cells were incubated for 24 h in DMEM/10% FBS/1%PenStrep media supplemented with no additional sugar, 50 mM sucrose or 50 mM lactose. Tissue culture grade 24-well plates were coated overnight with an ECM model mixture containing $0.5 \, \mu g/cm^2$ laminin from human fibroblasts (Sigma), $2.5 \, \mu g/cm^2$ fibronectin from human plasma (Sigma), and $6 \, \mu g/cm^2$ collagen I from rat tail (Sigma). Wells were blocked for 30 min with 1% BSA to block nonspecific binding and washed 3× with serum-free DMEM prior to addition of cells. Cells were harvested using 1 mM EDTA, washed 2× with serum-free DMEM and incubated with or without 10 mM EDTA (Thermo), to inhibit integrin activity, for 30 min in serum-free DMEM. Cells were added in the presence or absence of 10 mM EDTA in serum-free DMEM with no additional sugar, 10 mM sucrose or 10 mM lactose to ECM coated plates and incubated for 60 min at 37 °C. At end of incubation plates were placed on ice and processed for western blotting as described below.

**Immunoprecipitation**. Antibodies and concentrations used for immunoprecipitations are cataloged in Supplementary Table 2. Cells were lysed in 2% NP-40 buffer/Buffer A (150 mM NaCl, 0.5 mM Tris, 1 mM EDTA). Lysates were quantified using Bradford reagent. IGF1R, Integrins α4, β1, or β3 were immunoprecipated from at least 200 μg lysate mixed with 40 μl of BSA-blocked protein G-agarose bead slurry (#15920-010, Life Technologies) loaded with 2 μg anti-IGF1R (Abcam), anti-Integrins α4, β1 or β3 antibodies (Cell Signaling). Immunprecipitation was carried out overnight on a rotator at 4 °C, followed by washing with lysis buffer, elution by boiling in Laemmli reducing running buffer and western blot with LEA-biotin (Vector) lectin (for detection of I-branched glycans) or IGF1R, Integrins α4, β1 or β3 antibodies. As controls, immunprecipitations were performed in parallel with equal amounts of respective isotype control antibody.

**Western blot analysis**. Antibodies and concentrations used for western blot assays are cataloged in Supplementary Table 2. Cells were lysed in RIPA Buffer (#8990, ThermoFisher) in the presence of protease/phosphatase inhibitors (#88668, ThermoFisher) for 15 min on ice. Lysates were spun at 4 °C for 15 min and protein concentrations were determined using the BCA protein assay kit (#23227, ThermoFisher) according to the manufacturer's protocol. Equal amounts of total protein were separated on reducing 4–20% SDS-PAGE gradient gels (Bio-Rad) and transferred to nitrocellulose membranes, 0.2 μm (#162-0112, Bio-Rad). Membranes were blocked in Odyssey Blocking Buffer (#927-50000, Li-Cor) for 1 h at room temperature and then incubated with primary antibodies in Odyssey Blocking Buffer + Tris-buffered saline (TBS)/0.1% Tween-20 overnight at 4 °C. Subsequently, membranes were washed in TBS/0.1% Tween-20 and stained with IRDye-conjugated secondary antibodies (1:15,000 dilution) in Odyssey Blocking Buffer + Tris-buffered saline (TBS)/0.1% Tween-20 for 30 min at room temperature. Blots were washed in TBS/0.1% Tween-20 and analyzed on a Li-Cor Odyssey CLx Imaging System. Band intensities were calculated using ImageStudio software (Li-Cor). Full uncropped blots are available in Supplementary information.

**IGF-1 and RGD binding assays**. Cells were harvested in 1 mM EDTA, washed with DMEM/serum-free media and plated into 96-well round bottom tissue culture plates at $5 \times 10^5$ cells/well in 50 μl. Cells were serum-starved at 37 °C for 1 h. After incubation cells were placed immediately on ice and incubated with 10, 100, or 500 μg/ml of rhIGF1 conjugated to IRDye800CW (Custom order, Li-Cor) or with 0.5, 1, or 2 μM RGD conjugated to IRDye800CW (#926-09889, Li-Cor) for 1 h. Cells were washed with 2× with 1%BSA/PBS at 4 °C. Cells were fixed with 3.7% paraformaldehyde in PBS for 20 min at RT. Fixed cells were washed 3× with 0.1% Triton-X100. Cells were then blocked in Odyssey Blocking Buffer (#927-50000, Li-Cor) + 0.1% Tween-20 for 1 h at RT on rotator. Cells were incubated according to manufacturer's protocol with CellTag700 Stain (#926-41090, Li-Cor) for 1 h at RT on rotator. Cells were washed 3× with 0.1% Tween-20/PBS then analyzed on a Li-Cor Odyssey CLx Imaging System. Binding intensities were calculated using ImageStudio software (Li-Cor).

**Statistical analysis**. Statistical analyses were performed using Prism 7.0 software (GraphPad). For tests involving two groups, testing was carried out using unpaired two-tailed Student's $t$ test. When more than two groups were compared, a one-way analysis of variance (ANOVA) followed by Dunnett's multiple comparisons tests were performed. A two-way ANOVA followed by either Bonferroni's (two groups) or Dunnett's (more than two groups) multiple comparisons tests were used in cases where more than two groups were compared with repeated measures (i.e., in vivo tumor growth). For correlation of I-branch glycan expression and GCNT2 gene expression, linear regression was used. Statistical analysis for GlycoV4 microarray were done with limma[62], and Benjamini–Hochberg correction[63] was used to adjust the p values. Only genes with absolute fold change >2 and adjusted p value < 0.05 were considered differentially expressed. Based on statistical significance assessments in prior published data sets on the role of glycomics and cancer cell biology[11,21], we performed all in vitro and in vivo assessments a minimum of three times, unless otherwise noted. Data are presented as the means ± SEM. P values < 0.05 was were considered significant.

**Data availability**. The data generated in this paper has been deposited in the Gene Expression Omnibus (GEO) under accession number GEO: GSE94837.The following melanoma GEO datasets were used for gene expression analysis GSE31879[25], GSE4840[26], GSE44660[27], GSE7553[29], GSE15605[30], and GSE4587[31].

Whereever possible, all data have been made available to the public through accession codes, unique identifiers, or web links, and raw data associated with figures are included in the Supplementary Information section. All other remaining data are available within the Article and Supplementary Files, or available from the authors upon request.

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

## Acknowledgments

We thank Dr. Steven Barthel for his helpful discussions in the construction of gene expression vectors and cloning experiments. We also thank Dr. Hideo Takahashi for preparing the anti-i (OSK-28) and anti-I (OSK-14) monoclonal antibodies. This work was supported by the following grants NIH/NCI R01CA173610 and NIH/NIAID R21AI125476 (C.J. Dimitroff), NIH Predoctoral Fellowship Award F31CA171520 (J. Geddes Sweeney), Wellcome Trust 082098 (S.M. Haslam), Biotechnology and Biological Sciences Research Council BBF0083091 (A. Dell and S.M. Haslam), and AAI Career in Immunology Fellowship (N. Giovannone).

## Author contributions

Conceptualization: J.G.S., H.W., and C.J.D.; Methodology: J.G.S., J.L., N.G., and H.W.; Validation: J.L. and S.L.K.; Formal analysis: J.G.S., S.K., T.S.M., S.R.H., D.B., G.F.M., and H.W.; Investigation: J.G.S., J.L., A.A., S.R.H., S.L.K., and S.M.H.; Resources: Y.T. and

A.D.; Writing—original draft: J.G.S.; Writing—review and editing: J.G.S., N.G., and C.J. D.; Visualization: J.G.S., A.A., S.K., T.S.M., S.R.H., G.F.M., and S.M.H.; Supervision: C.J. D.; Funding acquisition: C.J.D.

## Additional information

**Competing interests:** The authors declare no competing interests.

