## [Peer Review File · Nature Communications]

Reviewers' Comments:

Reviewer #1:

Remarks to the Author:

Geddes-Sweeney et. al

Loss of GCNT2/I-branched glycans enhances growth factor receptor and adhesion mediated melanoma growth and survival pathways

This manuscript demonstrates that melanoma cell lines have reduced expression of glycosyltransferase GCNT2 compared to normal melanocytes and is associated with reduced I-branched glycans. In addition, authors show that GCNT2 levels are reduced in melanoma tissue using IHC and is correlates with aggressive metastatic disease. Authors also show that reducing GCNT2 expression with RNAi leads to increase in melanoma tumor growth in vivo and conversely overexpressing GCNT2 blocks tumor growth in vivo. Authors show that GCNT2 expression has no effect on melanoma cell growth but see difference when treated with IGF-1. Authors also detect changes in BCL-2 family members under conditions when plated on ECM. Lastly, authors show that IGF1R signaling is altered in conditions when GCNT2 including phosphorylation of IGF1R and AKT as well as integrin associated signaling such as FAK. Thus, authors conclude that melanoma cells reduce GCNT2 expression and reduce I-branched glycans which leads to increased growth specifically through IGF1R and integrin-ECM mediated signaling pathways.

Overall, the manuscript provides some novel data regarding role of GCNT2 and I-branched glycans as regulator of melanoma growth in vitro and in vivo. Major issue is the connection of GCNT2/I-branched glycans to IGF1R and integrins is weak and some of the conclusions/model is not supported by the data.

Major issues:

1. Weakest aspect of paper is connection of GCNT2 to IGF1R and integrin/ECM signaling. Authors need to show that I-branched glycans on IGF1R or a specific integrin is altered in cell lines in which GCNT2 is altered. In addition, aggressive melanoma cells should also contain IGF1R with increased I-branched glycans. Current data presented is correlative and not very convincing (especially for integrin signaling).
2. Figure 3 is critical for manuscript. Authors need to verify that antibody used for IHC to detect GCNT2 is specific. Authors should show that GCNT2 antibody does not detect GCNT2 in melanoma cells containing GCNT2 RNAi (this would also help results from Fig. 4 as authors don't show GCNT2 protein levels in RNAi expressing cells).
3. The change in apoptotic factors is not well tied to rest of manuscript since author's don't show any effect of GCNT2 on apoptosis in vitro or in vivo (authors could examine effects of GCNT2 on anchorage-independent growth or anoikis-resistance).
- 4.

Minor issues:

1. Some conclusions need to be toned down. For example page 6 line 167 "loss of GCNT2/I-branched glycans is a hallmark of melanoma transformation."
2. Author should explain results in Fig. 4F. Why do I-linear glycans increase by just overexpression of vector control?
3. Figs. 5A-C don't add much and should be included in supplemental.
4. Authors need to show changes in protein levels in Fig. 5H and 5I (RNA should be moved to supplemental).

Reviewer #2:

Remarks to the Author:

In this manuscript the investigators describe differences in the glycosylation gene-signature in melanomas compared to normal human melanocytes, in particular downregulation of the glycosyltransferase, GCNT2, leading to a loss of I-branched glycans. The authors performed GCNT2 knockdown and overexpression studies and show that loss of GCNT2 increased melanoma xenograft growth while presence of GCNT2 decreased melanoma growth. They also attempted to

identify the target by Western blotting with phosphorylated IGF1R, AKT, FAK and ERK to see if their presence is in agreement with the effect on growth response.

The manuscript is well written and the literature is well reviewed. The data showing changes in branched glycan are nicely demonstrated. However, the effects on in vitro cell proliferation and in vivo tumor growth, although statistically significant, are very small, ranging between 1.2-1.4 fold, and are unlikely to have clinical significance. Furthermore, changes in the phosphorylation status of IGF1R, AKT, FAK and ERK in response to GCNT2 KD or OE are also miniscule and hardly visible when compared to controls.

It is not clear why the investigators targeted the shRNA studies consistently using UACC62 melanoma cells and the overexpression studies A375. It is routine to use the same cell line for various manipulations.

Altogether, the investigators show changes in I-branched glycans in melanomas compared to normal melanocytes but failed to demonstrate significant functional responses to variation in GCNT2 expression.

Reviewer #3:

Remarks to the Author:

This is an interesting study from an established glycobiology group. In the study, the authors use a combination of data mining and experimentation to correlate loss of GCNT2 expression and melanoma progression. They demonstrate that this corresponds to loss of I-branched glycans on the cell surface, and interestingly results in increased linear i-glycans. Through knockdown and overexpression experiments they suggest a correlation between GCNT2 expression and xenograft tumor volume, in vitro growth rate, and growth factor and integrin signaling.

However, the paper has several major weaknesses. The novelty of the study lies in uncovering the connection between melanoma and GCNT2, a beta-1,6-GlcNAc transferase which is understudied. However, the study itself is almost purely correlative and lacks mechanistic insight.

Among the major issues not addressed is whether it is the loss of branched I-glycans or rather the increase in linear i-glycans that is the causal change. This could be sorted out by knocking down B3GNT enzyme(s) in melanoma cells (to reverse the increase in i-glycans in GCNT2 depleted conditions) and/or over-expressing B3GNT enzymes in GCNT2 replete conditions.

The authors then make an unjustifiably large leap from changes in cell surface glycosylation to changes in signaling, while ignoring the mechanistic details of the intervening steps. For example, there are proven galectin mediated mechanisms linking cell surface N-acetyllactosamine density (which is increased by linear i-glycans) to cell surface receptor maintenance and clustering. At the very least, the authors should investigate the effect of GCNT2 KD/OE on galectin binding and associated ability to control receptor clustering and endocytosis. Furthermore, the authors demonstrate in several instances that total protein levels of relevant proteins are unchanged. Yet the expectation is that changes in glycosylation are much more likely to affect cell surface loss of receptors by endocytosis and/or clustering rather than whole cell protein levels. Thus it is particularly important to assess for changes in cell surface levels and ligand induced clustering of the integrins and IGF1R.

Finally, the authors do not directly address whether the IGF-1 and integrin signaling pathways are indeed the critical mediators in aberrantly glycosylated melanoma or just two of many affected pathways. This is particularly critical given the very modest differences observed in phosphorylation within these pathways. The assertion that these are important pathways driving melanoma progression requires demonstration of reversal, preferably in vivo.

Minor issues include:

It is not clear if the statistical analyses employed included correction for multiple comparisons (e.g. Fig 4a-c). Multiple comparison corrections should be performed where appropriate and should be indicated in the corresponding individual figure legends.

Since generation of stable over-expressor lines involves insertion into the genome, more than one over-expressor line should be evaluated to rule-out effects from gene insertion.

At the bottom of page 5 (lines 159-163), the authors state that through use of Kifunensine and PPPP they “confirmed that I-branched glycans are present on N-linked glycoproteins as well as on glycolipids on melanoma cells (Supplementary Fig. 1c-e).” However, in these figures they only show a combination treatment of kifunensine and PPPP. Based on this data, the conclusion is unfounded. They should show treatment with Kifunensine and PPPP individually in order to demonstrate the relative contribution of N-glycosylated proteins and glycolipids to cell surface I-branched glycans.

At the bottom of page 6 (lines 182-185), the authors state “data mining of publicly available datasets...further confirmed lower GCNT2 levels...(fig 3g).” However the data shows significant differences in only two out of three data sets. This should be explicitly stated in the text.

On page 7 (lines 192-196). KD and OE acronyms should be defined in their first instance. This sentence needs to be rewritten for clarity (or perhaps just a typo and needs to be two sentences rather than one).

With regard to the immunohistochemistry (fig3), it is unclear whether staining intensity scores were done blinded or not. Perhaps this is not feasible due to the nature of the specimens, but this should be mentioned in the text.

Figure 2a: Several typos present. 1) NAGK is not a nucleotide sugar transporter. 2) Several instances of NUCLEAR sugar transporter which should say NUCLEOTIDE sugar transporter

Point-by-Point Responses to the Referees' Comments:

REVIEWER #1

Point 1.) The reviewer states: “*The weakest aspect of paper is connection of GCNT2 to IGF1R and integrin/ECM signaling. Authors need to show that I-branched glycans on IGF1R or a specific integrin is altered in cell lines in which GCNT2 is altered. In addition, aggressive melanoma cells should also contain IGF1R with increased I-branched glycans. Current data presented is correlative and not very convincing (especially for integrin signaling).*”

Our Response. We agree with the reviewer that it is essential to demonstrate I-branched glycans specifically on IGF1R and on specific integrin subunits. To address this question and because anti-human I-antigen (Osk-14) IgM antibody does not function in the context of Western blotting or in immunoprecipitation (data not shown), we utilized LEA lectin. We now show in **Supplemental Figure 4 panel k** that LEA lectin binds preferentially to melanoma cells expressing GCNT2/I-branched glycans, indicating that LEA favors binding to I-branched glycans over i-linear glycans. We immunoprecipitated IGF1R and integrin $\alpha 4$, integrin $\beta 1$ or integrin $\beta 3$ from both UACC62 scrambled control and GCNT2 KD and from A375 vector control and GCNT2 OE cell variants. We then ran the immunoprecipitated receptors from each cell variant on a gradient 4-20% SDS-PAGE gel and Western blotted with LEA lectin. We now show in **Figure 5a-d** that IGF1R and several integrin subunits isolated from melanoma cells expressing GCNT2 exhibited higher LEA lectin reactivity than controls, indicating that IGF1R and $\alpha 4$, $\beta 1$ and $\beta 3$ integrins are modified with I-branched glycans in cell lines with high GCNT2 levels. This is also reflected in the text on **Page 9 (lines 240-242)**. These new findings help support our conclusions that GCNT2/I-branched glycans interfere with IGF1R- and integrin-ECM-mediated cell growth and survival signaling pathways in melanoma cells.

Point 2.) The reviewer states: “*Figure 3 is critical for manuscript. Authors need to verify that antibody used for IHC to detect GCNT2 is specific. Authors should show that GCNT2 antibody does not detect GCNT2 in melanoma cells containing GCNT2 RNAi (this would also help results from Fig. 4 as authors don't show GCNT2 protein levels in RNAi expressing cells).*”

Our Response. We agree with this recommendation and have added additional assessments to validate GCNT2 protein detection by IHC. We now present new IHC data in **Supplemental Figure 3 panel d** showing GCNT2 staining in UACC62 scrambled control and little/no GCNT2 staining in UACC62 GCNT2 KD cell variants. Additionally, in **Supplemental Figure 3 panel e**, we found that A375 vector control cells had low GCNT2 staining and A375 GCNT2 OE cells stained strongly with GCNT2 Ab. This is also reflected in the text on **Page 7 (lines 195-197)**. This new data now confirm knockdown and overexpression of GCNT2 protein in our melanoma cell variants and also validates the specificity of the GCNT2 Ab used for IHC in clinical melanoma specimens.

Point 3.) The reviewer states: “*The change in apoptotic factors is not well tied to rest of manuscript since author's don't show any effect of GCNT2 on apoptosis in vitro or in vivo (authors could examine effects of GCNT2 on anchorage-independent growth or anoikis-resistance).*”

Our Response. We thank the reviewer for this great suggestion and have added additional experiments on the role of GCNT2 in melanoma cell tumorigenicity/survival in anchorage independent growth. We plated single melanoma cells in 1% methylcellulose in DMEM/10%FBS/1%PenStrep and allowed cell colonies to form over the course of 12 days (re-

feeding colonies every 3 days with fresh media). On day 12, we quantified the number of colonies and harvested the cells to analyze cell death by flow cytometry and for RNA analysis of pro-survival and pro-apoptotic factors. We found in melanoma cells expressing GCNT2/I-branched glycans smaller and fewer colonies formed, which corresponded to an increase in dead cells, compared to GCNT2 low melanoma cells. This new data is now represented along with our *in vivo* growth data in revised **Figure 4 panels e-j** and is represented in the text on **Page 8 (lines 213-220)**.

Minor Point 1.) The reviewer states: “*Some conclusions need to be toned down. For example page 6 line 167 “loss of GCNT2/I-branched glycans is a hallmark of melanoma transformation.”*”

Our Response. We agree with this recommendation and have gone through the manuscript to revise some of our conclusions so that they are more supported by the data presented. For the particular example mentioned by the reviewer, we have removed that conclusion from the manuscript and we now conclude this section with “Taken together, these data indicate that GCNT2 expression is inversely correlated with melanoma progression and highlights its potential use as a biomarker that correlates with stage of disease and associated virulence.” This can be found on **Page 7 (lines 185-188)**.

Minor Point 2.) The reviewer states: “*Author should explain results in Fig. 4F. Why do I-linear glycans increase by just overexpression of vector control?*”

Our Response. A375 vector control cells have low levels of GCNT2 (similar to A375 WT melanoma cells) and thus, primarily display i-linear glycans on the cell surface. When we overexpress GCNT2 in this cell line (A375 GCNT2 OE cells), GCNT2 modifies those i-linear glycans with I-branched glycans and therefore decreases the amount of i-linear glycans as detected by the OSK-28 antibody and increases the amount of I-branched glycans on the cell surface as detected by the OSK-14 antibody. Note that these data have now been moved to **Supplemental Figure 4 panels d-f**.

Minor Point 3.) The reviewer states: “*Figs. 5A-C don’t add much and should be included in supplemental.*”

Our Response. We agree with this recommendation and have moved these figure panels to **Supplemental Figure 5 panels a-c**.

Minor Point 4.) The reviewer states: “*Authors need to show changes in protein levels in Fig. 5H and 5I (RNA should be moved to supplemental).*”

Our Response. We agree with the reviewer and have now run western blots of lysates from UACC62 scrambled control and GCNT2 KD and A375 vector control and GCNT2 OE melanoma cells plated on an extracellular matrix (ECM) layer for 48hrs. Consistent with our RNA findings, we found that UACC62 scrambled cells had higher pro-apoptotic Bid protein levels and lower pro-survival BCL-XL protein levels than GCNT2 KD cells when plated on ECM. Conversely, A375 vector control cells had lower pro-apoptotic Bax protein levels and higher pro-survival BCL-XL protein levels than GCNT2 OE cells when plated on ECM. This strengthens our data that GCNT2 regulates melanoma cell survival and this new data is now presented in **Figure 5 panels h and i** and in the text on **Pages 9 and 10 (lines 252-272)**.

REVIEWER #2

Point 1.) The reviewer states: “*The data showing changes in branched glycan are nicely demonstrated. However, the effects on in vitro cell proliferation and in vivo tumor growth, although statistically significant, are very small, ranging between 1.2-1.4 fold, and are unlikely to have clinical significance. Furthermore, changes in the phosphorylation status of IGF1R, AKT, FAK and ERK in response to GCNT2 KD or OE are also miniscule and hardly visible when compared to controls.*”

Our Response. We understand this comment to some degree, though our data did reach a high level of statistical significance. Importantly, changes in GCNT2 expression in primary and metastatic melanoma specimens showed statistically significant downregulation of GCNT2 as melanomas progress. Clinically, a major advance of this paper is the potential use of GCNT2 as an inverse correlate (biomarker) to help predict which patients are at risk for progression to metastatic disease.

Published data in field of glycobiology are often underappreciated albeit statistically significant due to lack of large experimental differences when compared with controls, making the results appear insignificant to the pathogenesis of that particular disease. But it is important to remember that glycans have vast structural complexity and heterogeneity and the numerous functions of glycans are based on their structural diversity. In most instances glycans “tune” a primary function of a protein rather than turning it on or off. Thus, it is the ‘net’ effect (both positive/negative regulation) of the glycan on a distinct membrane protein function that is assayed in a biological response.

To further strengthen our data on the role of GCNT2/I-branched glycans in melanoma growth and survival, we now include additional tumorigenicity and survival experimental data. First, we plated single GCNT2 KD, GCNT2 OE or control cell variants in 1% methylcellulose in DMEM/10%FBS/1%PenStrep for 12 days. On Day 12, we quantified the number of colonies and assessed cell death by FACS and RNA analysis. We found that melanoma cell expressing GCNT2/I-branched glycans formed significantly fewer colonies, which corresponded with an increase in cell death. These new data are now presented in **Figure 4 panels e-f.**

We also repeated our *in vivo* tumor xenograft studies. On Day 24, we harvested the tumors and analyzed protein and gene expression of various pro-apoptotic and pro-survival BCL-2 family members. Similar to our new *in vitro* tumorigenicity and survival studies, we found that xenograft tumors expressing GCNT2/I-branched glycans expressed significantly more pro-apoptotic Bid or Bax protein and gene expression and significantly less pro-survival BCL-XL protein and gene expression. This new data are now presented in **Supplemental Figure 7 panels e-p.**

In all, we believe that GCNT2 modifies N-glycans on multiple cell surface glycoproteins and though there are modest differences in the individual signaling pathways affected by GCNT2, the combine differences lead to a significant decrease in melanoma cell growth and survival *in vivo*.

Point 2.) The reviewer states: “*It is not clear why the investigators targeted the shRNA studies consistently using UACC62 melanoma cells and the overexpression studies A375. It is routine to use the same cell line for various manipulations.*”

Our Response. We understand why the reviewer questions our use of different cell lines for our GCNT2 functional assessments. Because GCNT2 encodes an enzyme, we found that when we overexpressed the GCNT2 gene in UACC62 cells, ***which already have high native expression of GCNT2***, the enzymatic activity of GCNT2 in the overexpression (OE) cell line was not enhanced, as we did not detect a significant increase in the amount of I-branched glycans on the cell surface between our control and GCNT2 OE cell variants (**Please see Figure A panels a and b below.**) We believe that this may be because all available galactose substrates on N-glycans are already

being modified with I-branches in UACC62 vector control cells. Therefore, when we overexpress GCNT2 in these cells we do not observe an increase in I-branched glycans.

Figure A. GCNT2 overexpression does not increase I-branched glycans on UACC62 melanoma cells.

a) qRT-PCR of GCNT2 gene expression in UACC62 vector control and GCNT2-overexpressing (OE) cell variants.

b) Representative flow cytometry plot (left) and quantitation (right) of cell surface I-branched glycan expression on UACC62 vector control and GCNT2 OE cell variants.

Thus, we decided to use A375 melanoma cells, which have natively low levels of GCNT2 and I-branched glycans on the cell surface (**Supplemental Figure 2 panels a and b**), to overexpress GCNT2 and demonstrate increases in I-branched glycan formation. Conversely, we used UACC62 melanoma cells, which have natively high expression of GCNT2 and I-branched glycans on the cell surface (**Supplemental Figure 2 panels a and b**), to knockdown GCNT2 and decrease I-branched glycans on the cell surface. We agree that using different transduced cell lines in bioassays may make it difficult to assess opposing effects in bioassays. However, we do see opposing effects using different cell lines, which we believe reinforce the importance of GCNT2/I-branches in melanoma cell activity, independent of the genetically heterogeneous nature of different melanoma cells lines.

REVIEWER #3

Major Point 1.) The reviewer states: “Among the major issues not addressed is whether it is the loss of branched I-glycans or rather the increase in linear i-glycans that is the causal change. This could be sorted out by knocking down B3GNT enzyme(s) in melanoma cells (to reverse the increase in i-glycans in GCNT2 depleted conditions) and/or over-expressing B3GNT enzymes in GCNT2 replete conditions.”

Our Response. We understand the reviewer’s comment that we do not know whether it is the loss of I-branched glycans or rather the increase in i-linear glycans that is the causal change regulating melanoma growth and survival.

To address this comment, we mined public databases comparing gene expression in melanocytes versus melanomas for expression differences in various β 3GNT genes. We did not find any significant changes in expression of the major β 3GNT genes involved in extending polyLacNAc glycans (data not shown). Moreover, in our glycomic microarray analysis (**Figure 2 panel d**) comparing NHEMs versus A375 and G361 melanoma cell lines, we actually found that NHEMs had higher expression of β 3GNT9 compared to melanoma cells, yet NHEMs also have more GCNT2 corresponding to high expression of I-branched glycans and lower expression of i-linear glycans. Nevertheless, in depth analysis of our mass spectrometry data also reveals that

normal melanocytes do have extended polyLacNAcs, consisting of 4 or more LacNAc residues, which may or may not be modified with I-branched glycans (**Supplementary Fig 1a-c - peaks 4939, 5388 and 5837**). With this, we hypothesize that extended i-linear polyLacNAcs could potentially be regulated by β 3GNT and GCNT2 competition for the same donor sugar, UDP-GlcNAc, thereby limiting i-linear polyLacNAc length when GCNT2 is expressed. Overall, in our model, we believe that downregulation of GCNT2 is important for the loss of I-branched glycans in melanoma cells and the increase in extended polyLacNAcs. We have further addressed this possibility in the manuscript in the **Discussion** section on **Pages 14 and 15 (lines 383-406)**.

Major Point 2. The reviewer states: “*The authors then make an unjustifiably large leap from changes in cell surface glycosylation to changes in signaling, while ignoring the mechanistic details of the intervening steps. For example, there are proven galectin mediated mechanisms linking cell surface N-acetylglucosamine density (which is increased by linear i-glycans) to cell surface receptor maintenance and clustering. At the very least, the authors should investigate the effect of GCNT2 KD/OE on galectin binding and associated ability to control receptor clustering and endocytosis. Furthermore, the authors demonstrate in several instances that total protein levels of relevant proteins are unchanged. Yet the expectation is that changes in glycosylation are much more likely to affect cell surface loss of receptors by endocytosis and/or clustering rather than whole cell protein levels. Thus it is particularly important to assess for changes in cell surface levels and ligand induced clustering of the integrins and IGF1R.*”

Our Response. We agree with the reviewer that our data do not address how differential GCNT2/I-branched glycan expression interferes with IGF1R and integrin-mediated growth and survival signaling. Because linear polyLacNAc glycans often serve as ligands for galectins and because GCNT2 modifies these linear glycans with I-branches, we were very interested in investigating the impact of GCNT2/I-branched glycans on galectin binding.

Interestingly, we found that GCNT2/I-branched glycans inhibited Galectin-3 (Gal-3) binding but had no effect on Galectin-1 (Gal-1) binding (**Please see Figure B panels a-d below**). With this, we treated GCNT2 KD, GCNT2 OE and control cell variants with recombinant human (rh)Gal-3 in the context of IGF-1 or ECM stimulation and ran western blots to assess downstream signaling. Unexpectedly, rhGal-3 did not affect IGF1R- or integrin-mediated signaling in GCNT2 KD or GCNT2 OE cell variants (**Please see Figure B panels g and h below**).

The reviewer also mentions that glycans have been shown to alter receptor levels on the cell surface to modulate cellular responses, typically through interactions with galectins. Therefore, we also studied the role of Gal-3 in mediating IGF1R internalization in response to stimulation with rhIGF-1 in GCNT2 KD, GCNT2 OE and control cell variants by flow cytometry. Similar to our signaling results, we found that rhGal-3 did not have any effect on internalization of IGF1R. Notably, rhGal-3 alone also did not induce internalization (**Please see Figure C panels a-h below**). In addition, at steady state, we do not see significant differences in cell surface receptor levels in i-linear and I-branched glycan expressing cells. These new data are now presented in the manuscript in **Supplemental Figure 6a-h**. We have also added a discussion point in the Discussion section on **Pages 17 and 18 (lines 462-468)**.

Thus, although GCNT2/I-branched glycans inhibit Gal-3 binding, we do not believe that these differences in Gal-3 binding are responsible for mediating the observed differential IGF1R or integrin-mediated signaling in GCNT2 KD or GCNT2 OE cell variants. However, we are exploring other areas of malignancy in which Gal-3 binding differences may play a role, but are outside the framework of this manuscript. To this end, is it possible to not present this data showing GCNT2/I-branch inhibition of Gal-3 binding in the online **Point-by-Point Response to Referees** document? We ask that this data remain confidential.

Figure B. I-branched glycans inhibit Gal-3 binding but does not alter IGF1:IGF1R or integrin:ECM mediated signaling

(a,b) Histogram and quantitation of Gal-1 (20ug/ml) binding to UACC62 control and GCNT2 KD **(a)** and A375 control and GCNT2 OE **(b)** cell variants.

(c,d) Histogram and quantitation of Gal-3 (10ug/ml) binding to UACC62 control and GCNT2 KD **(c)** and A375 control and GCNT2 OE **(d)** cell variants.

(e) Schematic of IGF-1 (100ng/ml) + rhGal-3 (10ug/ml) treatment.

(f) Schematic of ECM + rhGal-3 (10ug/ml) treatment.

(g) Immunoblot of phosphorylated and total IGF1R of UACC62 control and GCNT2 KD (top) and A375 control and GCNT2 OE (bottom) cell variants.

(h) Immunoblot of phosphorylated and total FAK of UACC62 control and GCNT2 KD (top) and A375 control and GCNT2 OE (bottom) cell variants.

Figure C. Gal-3 does not affect IGF-1 induced IGF1R internalization

(a) Histograms of IGF1R on cell surface of A375 vector control and GCNT2 OE cell variants after rhIGF-1 stimulation +/- 20µg/ml rhGal-3 +/- 50mM Lactose.

(b) Quantitation of percent of IGF1R internalized after rhIGF-1 stimulation +/- 20µg/ml rhGal-3 +/- 50mM Lactose in A375 vector control and GCNT2 OE cell variants.

(c) Histograms of IGF1R on cell surface of A375 vector control and GCNT2 OE cell variants after rhIGF-1 stimulation.

(d) Histograms of IGF1R on cell surface of A375 vector control and GCNT2 OE cell variants after treatment with 20µg/ml rhGal-3 +/- 50mM Lactose.

Continued Figure C

(e) Histograms of IGF1R on cell surface of UACC62 scrambled control and GCNT2 KD cell variants after rhIGF-1 stimulation +/- 20µg/ml rhGal-3 +/- 50mM Lactose.

(f) Quantitation of percent of IGF1R internalized after rhIGF-1 stimulation +/- 20µg/ml rhGal-3 +/- 50mM Lactose in UACC62 scrambled control and GCNT2 KD cell variants.

(g) Histograms of IGF1R on cell surface of UACC62 scrambled control and GCNT2 KD cell variants after rhIGF-1 stimulation.

(h) Histograms of IGF1R on cell surface of UACC62 scrambled control and GCNT2 KD cell variants after treatment with 20µg/ml rhGal-3 +/- 50mM Lactose.

To further explore how differential GCNT2/I-branched glycan expression interferes with IGF1R and integrin-mediated growth and survival signaling, we investigated the impact of I-branched glycans on ligand binding to the receptors. Intriguingly, the IGF-1 binding site of IGF1R, which has been mapped to residues 223-274 of human IGF1R α chain, contains one N-linked glycan site. Additionally, though the ligand binding sites for integrins are not as well-defined, both integrin α and β chains have multiple N-linked glycans sites located in the β -propeller repeat region of integrin α chains and the β A domain of integrin β chains, where ligands are likely to bind. Thus, we hypothesized that the addition of I-branches to N-linked glycans within ligand binding sites of the receptors may negatively regulate ligand binding to the receptor. To test this, we used the cognate ligands for IGF1R and fibronectin-binding integrins, IGF-1 and RGD peptide (arginine-glycine-aspartic acid motif in fibronectin), respectively, conjugated to near-infrared probe – IRDye800CW from LiCor Biosciences. We found that the presence of I-branched glycans inhibited both IGF-1 and RGD-peptide binding. Importantly, kifunensine-treatment of GCNT2 variant and control cells to normalize N-glycans showed no difference in IGF-1 or RGD-peptide binding. Together, these data indicate that the presence of GCNT2/I-branched glycans reduce ligand binding to IGF1R and fibronectin binding integrins. These new data are reflected in the manuscript in **Figure 7 panels a-h** and in the text on **Pages 12 and 13 (lines 325-345)**.

Major Point 3.) The reviewer states: “*The authors do not directly address whether the IGF-1 and integrin signaling pathways are indeed the critical mediators in aberrantly glycosylated melanoma or just two of many affected pathways. This is particularly critical given the very modest differences observed in phosphorylation within these pathways. The assertion that these are important pathways driving melanoma progression requires demonstration of reversal, preferably in vivo.*”

Our Response. We understand the reviewer’s comment on the implication of our work. We believe that GCNT2 modifies N-glycans on multiple cell surface glycoproteins and, though we focus on two major signaling pathways and related glycoprotein receptors, our results indicate that global cellular differences in GCNT2/I-branched glycan expression lead to a significant decrease in melanoma cell growth *in vivo*. We have now added additional discussion points regarding this, which can be found in the **Discussion** section on **Page 17 (lines 450-459)** and we have also modified our model in **Figure 8b** to reflect the likely presence of other glycoproteins modified by GCNT2/I-branches, which may also affect *in vivo* tumor growth.

Minor Point 1.) The reviewer states: “*It is not clear if the statistical analyses employed included correction for multiple comparisons (e.g. Fig 4a-c). Multiple comparison corrections should be*

performed where appropriate and should be indicated in the corresponding individual figure legends.”

Our Response. We understand the reviewer’s comment and our statistical analyses do include correction for multiple comparisons. We have modified the **Methods** section on **Page 31 (lines 814-820)** as follows: “Statistical analyses were performed using Prism 7.0 software (GraphPad). For tests involving two groups, testing was carried out using unpaired two-tailed Student’s *t* test. When more than two groups were compared, a one-way analysis of variance (ANOVA) followed by Dunnett’s multiple comparisons tests were performed. A two-way ANOVA followed by either Bonferroni’s (two groups) or Dunnett’s (more than two groups) multiple comparisons tests were used in cases where more than two groups were compared with repeated measures (i.e. *in vivo* tumor growth).”

Minor Point 2.) The reviewer states: “*Since generation of stable over-expressor lines involves insertion into the genome, more than one over-expressor line should be evaluated to rule-out effects from gene insertion.*”

Our Response. We agree that site of insertion can theoretically impact an observed phenotype. However, we do not believe this is the case in our A375 vector control and GCNT2 OE cell variants. When we treated A375 control and GCNT2 OE cell variants *in vitro* with kifunensine to normalize cell surface N-glycans to high mannose glycan structures, we did not observe any differences in IGF1R-AKT or FAK-ERK signaling between our control and GCNT2 OE cells (Please see **Figure 6 panels g-k**). This indicates that it is indeed the presence I-branched glycans on the cell surface and not gene insertion, mediating our observed differential signaling. If our observed affects were due to insertion into the genome, we believe that we would also have observed differences in signaling in our kifunensine treated cell variants.

Minor Point 3.) The reviewer states: “*At the bottom of page 5 (lines 159-163), the authors state that through use of Kifunensine and PPPP they “confirmed that I-branched glycans are present on N-linked glycoproteins as well as on glycolipids on melanoma cells (Supplementary Fig. 1c-e).” However, in these figures they only show a combination treatment of kifunensine and PPPP. Based on this data, the conclusion is unfounded. They should show treatment with Kifunensine and PPPP individually in order to demonstrate the relative contribution of N-glycosylated proteins and glycolipids to cell surface I-branched glycans.*”

Our Response. We agree and now include cells treated with kifunensine and PPPP individually in **Supplemental Figure 2 panels c-f**. Interestingly, when we inhibited complex N-glycan formation on the cell surface with kifunensine, we actually detected more I-branched glycans on the cell surface. When we inhibited glycolipid formation with PPPP, we did see a slight decrease in I-branched cell surface glycans, though it did not account for total I-branched glycans and did not reach significance. With this, we had two potential hypotheses: 1.) The observed increase in I-branched glycans in kifunensine-treated cells was due to the reduction of the presence of highly complex N-glycans on glycoproteins that normally mask I-branches on glycolipids from detection by the OSK-14 (I-branched glycan specific) antibody and 2.) GCNT2 is able to modify both N-linked glycans and glycolipids in melanoma cells. Therefore, when we inhibited both complex N-glycan formation with kifunensine and glycolipid formation with PPPP, we completely eliminated detection of I-branched glycans on the cell surface, indicating that GCNT2 adds I-branches to N-linked glycans on glycoproteins and to glycolipids.

Minor Point 4.) The reviewer states: “At the bottom of page 6 (lines 182-185), the authors state “data mining of publicly available datasets...further confirmed lower GCNT2 levels...(fig 3g).” However the data shows significant differences in only two out of three data sets. This should be explicitly stated in the text.”

Our Response. We agree with the reviewer and have modified the text on **Page 7 (lines 183-185)** and now state, “In addition, data mining further confirmed lower GCNT2 mRNA levels in clinical metastatic melanoma specimens compared to primary melanoma specimens from two of three independent clinical cohorts.”

Minor Point 5.) The reviewer states: “On page 7 (lines 192-196). KD and OE acronyms should be defined in their first instance. This sentence needs to be rewritten for clarity (or perhaps just a typo and needs to be two sentences rather than one).”

Our Response. We agree and now specifically define KD and OE acronyms on **Page 7 (lines 192-195)**.

Minor Point 6.) The reviewer states: “With regard to the immunohistochemistry (fig3), it is unclear whether staining intensity scores were done blinded or not. Perhaps this is not feasible due to the nature of the specimens, but this should be mentioned in the text.”

Our Response. A third party technician, who did not know or have information on tumor grade/stage or site of metastasis, scored the IHC assessments under the guidance of a pathologist. We now explicitly state this in the **Methods on Page 27 (lines 702-705)**.

Point 10.) The reviewer states: “Figure 2a: Several typos present. 1) NAGK is not a nucleotide sugar transporter. 2) Several instances of NUCLEAR sugar transporter which should say NUCLEOTIDE sugar transporter.”

Our Response. We thank the reviewer for their help with typos and we have made the following changes (indicated below by red text and underlining).

B3GALNT1 (Glycosyltransferase)

ST3GAL5 (Glycosyltransferase)

SLC35B3 (Nucleotide Sugar Transporter)

NAGK (Nucleotide Synthesis)

SLC35F2 (Nucleotide Sugar Transporter)

HS6ST2 (Glycosyltransferase)

These changes are now reflected in **Figure 2 panel d**.

Reviewers' Comments:

Reviewer #1:

Remarks to the Author:

Authors have addressed all major and minor issues.

One minor edit is recommended before publication.

Page 8 line 213. Authors describe data shown in Fig. 4e,f as "a three-dimensional culture system". The assay itself seems to be an anchorage-independent assay using methylcellulose. Authors should label it as such in the text to make it clear to readers (since this is not a 3D culture system).

Reviewer #2:

Remarks to the Author:

The authors performed additional studies and addressed all the criticism raised by the reviewers.

Reviewer #3:

Remarks to the Author:

The authors have failed to address all major issues that were raised in my original review. They also incorrectly claim that their data demonstrates I-branching reduces ligand binding (Fig. 7). Rather the data is more supportive of reduced receptor number (see discussion below in Major Point two). We are left with a manuscript that describes correlations, lacks mechanistic insight and has mis-interpretation of data leading to incorrect conclusions.

Major Point 1: It was requested that the authors knockdown and/or over-express B3GnT enzymes to assess whether gain of linear polyLacNAc versus loss I-branching is the critical physiological change. This was not done.

Major Point 2: The authors were requested to investigate surface expression/endocytosis/clustering of IGF1R and integrins along with alterations in binding to galectin. The authors provide data in Supplemental Fig 6 and in the rebuttal to reviewers (that they do not put in the manuscript) that GCNT2/I-branched glycans inhibit Galectin-3 but not Galactin-1 binding, yet then claim that this does not alter signaling or surface levels/endocytosis of IGF1R. This response is not convincing and has multiple issues:

i) The authors claim that GCNT2/I-branched glycans do not alter IGF1R or integrin steady state surface levels. This is not supported by the data. First, Figure Ce,g,h all clearly show that knockdown (KD) of GNCNT2 reduces surface levels of IGF1R, which is in conflict with the single profile shown in Supplemental Figure 6b. Second, Supplemental Figure 6d shows that over-expression (OE) of GNCNT2 reduces surface levels of IGF1R, which is consistent with reduced ligand binding in Figure 7a. Third, KD of GNCNT2 appears to increase surface levels of Integrin $\alpha 6$ and $\beta 1$, while OE has the opposite effect (Supplemental Figure 6f,h). The latter is consistent with reduced ligand binding in Fig. 7b. Fourth, MFI are not provided for any of the data, which appears to have been done only once. These flow cytometry experiments must be done with triplicate staining and a titration of antibody and be presented as mean with error bars in addition to the histograms.

ii) The authors incorrectly claim that Figure 7 demonstrates altered ligand binding. MFI of ligand binding to cells depends on both affinity and receptor number, which this data does not distinguish. Scatchard analysis and saturation binding is required to distinguish receptor number vs affinity in this type of binding experiment. Alternatively, BiaCore could be used to directly assess affinity. Given the points raised above, Figure 7 is likely a result of altered receptor number rather than a change in affinity.

- iii) There are 14 different mammalian galectins and only two were examined. Other galectins may be involved such as Galectin-9, but this is not explored.
- iv) Galectin binding also regulates clustering of receptors in the plane of the membrane to affect signaling, a mechanism distinct from regulation of endocytosis. This mechanism is particularly relevant for integrin signaling, yet this was not examined as requested.
- v) Adding exogenous galectin is a poor approach as it does not consider the presence of endogenous galectins. Moreover, only a single dose was examined, which may be supra-physiological and induce changes not representative. A titration is necessary, as is knockdown of endogenous galectin or use of lactose in the absence of exogenous galectin (excess exogenous galectin may prevent disruption of endogenous galectins by lactose).
- vi) The data in Figure C have several technical issues that need to be addressed before conclusions can be drawn. Critical controls are not done side-by-side. For example, in Figure C a,b,e,f, addition of IGF1 alone needs to be compared directly to IGF1 +galectin-3 in both vector control and the GNCNT2 OE and KD. Second, the endocytosis assay employed in Figure C b,f is not described and there are no error bars. Third, as noted above, experiments need to be done with triplicate staining and presented as mean with error bars.

Major Point 3: The authors were asked to demonstrate that the very small effects on IGF1R and integrin signaling were relevant to the tumor phenotype by reversal. This was not done. Given the small changes in signaling, it is unlikely that these are critical pathways responsible for the tumor phenotype yet this is the claim made in the abstract and elsewhere. The data simply do not support this conclusion.

REVISION 2: POINT-BY-POINT RESPONSES TO REVIEWERS

REVIEWER #1

The reviewer states, “*Authors have addressed all major and minor issues.*”

Minor Point: The reviewer states that, “*One minor edit is recommended before publication. Page 8 line 213. Authors describe data shown in Fig. 4e,f as “a three-dimensional culture system”. The assay itself seems to be an anchorage-independent assay using methylcellulose. Authors should label it as such in the text to make it clear to readers (since this is not a 3D culture system).*”

Our Response: We have changed that description in the **Results** section of **Figure 4e and 4f** and we now state on **Page 8 lines 213-215**, “Furthermore, utilizing an anchorage independent growth assay designed to study the tumourigenicity and survival of malignant cells, we found that melanoma cells expressing GCNT2/I-branched glycans formed smaller and fewer colonies (Fig. 4e,f).”

REVIEWER #2

The reviewer states that, “*The authors performed additional studies and addressed all the criticism raised by the reviewers.*”

REVIEWER #3

Major Point 1: The reviewer states that, “*It was requested that the authors knockdown and/or over-express B3GnT enzymes to assess whether gain of linear polyLacNAc versus loss I-branching is the critical physiological change. This was not done.*”

Our Response. We understand the reviewer’s comment about assessing B3GNT enzymes to examine whether loss of I-branched or gain of i-linear polyLacNAc glycans is the causal change regulating melanoma growth and survival.

We performed gene set enrichment analysis from the NCI60 cancer cell line panel. We found that melanomas have high expression of multiple B3GNT genes (**Fig A, panel a**), all of which may contribute to polyLacNAc formation in melanoma cells. Furthermore, we also mined public databases comparing gene expression in melanocytes versus melanomas for expression differences in various B3GNT genes. We did not find any significant changes in expression of the major B3GNT genes involved in extending polyLacNAc glycans (**Fig A, panels b-d**). Moreover, in our glycomic microarray analysis (**Figure 2 panel d**) comparing NHEMs versus A375 and G361 melanoma cell lines, we actually found that NHEMs had *higher* expression of B3GNT9 compared to melanoma cells, yet NHEMs also have more GCNT2 corresponding to high expression of I-branched glycans and lower expression of i-linear glycans. Nevertheless, in depth analysis of our mass spectrometry data also reveals that normal melanocytes do have extended polyLacNAcs, consisting of 4 or more LacNAc residues, which may or may not be modified with I-branched glycans (**Supplementary Fig 1a-c - peaks 4939, 5388 and 5837**). With this, we hypothesize that extended i-linear polyLacNAcs could potentially be regulated by B3GNT and GCNT2 competition for the same donor sugar, UDP-GlcNAc, thereby limiting i-linear polyLacNAc length when GCNT2 is expressed. Considering that there are (9) B3GNT enzymes and our glycome gene expression data (and other published data sets) showed no differences in any B3GNT gene expression between normal melanocytes and melanoma cells, we hypothesize that B3GNTs do not play a major role in this cancer model. Overall, in our model, we believe that downregulation of GCNT2 is important for the loss of I-branched glycans in melanoma cells and the increase in extended polyLacNAcs. We

have further addressed this possibility in the manuscript in the **Discussion** section on **Pages 16 and 17 (lines 417-439)**.

Figure A. Expression of B3GNT genes are not significantly different between normal melanocytes and melanomas. (a) Heatmap of different B3GNT gene expression in melanomas versus eight other common malignancies. Generated using the Oncomine (www.oncomine.com) Compendia Cell Line panel. **(b-c)** Box plots illustrating no difference in B3GNT1 **(b)**, B3GNT2 **(c)** or B3GNT8 **(d)** gene expression between melanocytes and melanomas in multiple datasets. Mean is representative line inside the box.

Major Point 2: The reviewer states that, “The authors were requested to investigate surface expression/endocytosis/clustering of IGF1R and integrins along with alterations in binding to galectin. The authors provide data in Supplemental Fig 6 and in the rebuttal to reviewers (that they do not put in the manuscript) that GCNT2/I-branched glycans inhibit Galectin-3 but not Galactin-1 binding, yet then claim that this does not alter signaling or surface levels/endocytosis of IGF1R. This response is not convincing and has multiple issues:”

Point i: The authors claim that GCNT2/I-branched glycans do not alter IGF1R or integrin steady state surface levels. This is not supported by the data. First, Figure Ce,g,h all clearly show that knockdown (KD) of GNCNT2 reduces surface levels of IGF1R, which is in conflict with the single profile shown in Supplemental Figure 6b. Second, Supplemental Figure 6d shows that over-expression (OE) of GNCNT2 reduces surface levels of IGF1R, which is consistent with reduced ligand binding in Figure 7a. Third, KD of GNCNT2 appears to increase surface levels of Integrin $\alpha 6$ and $\beta 1$, while OE has the opposite effect (Supplemental Figure 6f,h). The latter is consistent with reduced ligand binding in Fig. 7b. Fourth, MFI are not provided for any of the data, which appears to have been done only once. These flow cytometry experiments must be done with triplicate staining and a titration of antibody and be presented as mean with error bars in addition to the histograms.

Our Response. We agree that FACS histograms do not always clearly represent receptor expression; and, while we do see variations in IGF1R and integrin receptor cell surface expression from experiment to experiment, there were no consistent statistically significant differences in expression in GCNT2 KD and GCNT2 OE and control melanoma cells. We now

provide both representative histograms and graphs of mean±SEM of MFIs from multiple experiments (**Supplemental Fig. 6a-d**). Of note, cell surface expression of integrin $\beta 3$ in one of the two GCNT2 KD melanoma cell variants ($p = 0.046$), as well as cell surface expression of integrin αV in both GCNT2 KD melanoma cell variants was significantly decreased compared to control cells. Nonetheless, we do not believe that these differences are due to GCNT2/I-branched or i-linear glycan expression, as we still observed these alterations in cell surface expression of integrin $\beta 3$ ($p = 0.061$) and integrin αV in kifunensine treated GCNT2 KD and control cells (**Supplemental Fig. 6s,t**).

In addition, we performed titrations (in triplicate) of two different IGF1R antibodies and we did not observe a significant difference in cell surface expression of IGF1R for either antibody or at any concentration used in GCNT2 KD and GCNT2 OE and control melanoma cells (**Figure B, panels a-d**). We also performed titrations (in triplicate) of different antibodies recognizing different integrin subunits in GCNT2 OE and control melanoma cells. We did not observe any significant differences in cell surface expression of any of the integrin subunits analyzed at any concentration (**Figure B, panels e-i**) in GCNT2 OE and control melanoma cells.

Figure B. GCNT2 cell variants express similar cell surface levels of IGF1R and various integrin α and β subunits. (a) Representative histograms (left) and quantitation (right) of cell surface IGF1R

Figure B Continued. expression using titration of IGF1R antibody (Biolegend) in UACC62 control and GCNT2 KD cell variants. **(b)** Representative histograms (left) and quantitation (right) of cell surface IGF1R expression using titration of IGF1R antibody (Abcam) in UACC62 control and GCNT2 KD cell variants. **(c)** Representative histograms (left) and quantitation (right) of cell surface IGF1R expression using titration of IGF1R antibody (Biolegend) in A375 control and GCNT2 OE cell variants. **(d)** Representative histograms (left) and quantitation (right) of cell surface IGF1R expression using titration of IGF1R antibody (Abcam) in A375 control and GCNT2 OE cell variants. **(e)** Representative histograms (left) and quantitation (right) of cell surface Integrin β 1 expression using titration of Integrin β 1 antibody in A375 control and GCNT2 OE cell variants. **(f)** Representative histograms (left) and quantitation (right) of cell surface Integrin β 3 expression using titration of Integrin β 3 antibody in A375 control and GCNT2 OE cell variants. **(g)** Representative histograms (left) and quantitation (right) of cell surface Integrin α 4 expression using titration of Integrin α 4 antibody in A375 control and GCNT2 OE cell variants. **(h)** Representative histograms (left) and quantitation (right) of cell surface Integrin α 6 expression using titration of Integrin α 6 antibody in A375 control and GCNT2 OE cell variants. **(i)** Representative histograms (left) and quantitation (right) of cell surface Integrin α V expression using titration of Integrin α V antibody in A375 control and GCNT2 OE cell variants. Results are representative of $n = 3$ experiments (mean \pm SEM); Statistical analyses were performed using Prism 7.0 software (GraphPad). For tests involving two groups, testing was carried out using unpaired two-tailed Student's t test. When more than two groups were compared, a one-way analysis of variance (ANOVA) followed by Dunnett's multiple comparisons tests were performed.

Point ii: *The authors incorrectly claim that Figure 7 demonstrates altered ligand binding. MFI of ligand binding to cells depends on both affinity and receptor number, which this data does not distinguish. Scatchard analysis and saturation binding is required to distinguish receptor number vs affinity in this type of binding experiment. Alternatively, BiaCore could be used to directly assess affinity. Given the points raised above, Figure 7 is likely a result of altered receptor number rather than a change in affinity.*

Our response. We agree with the reviewer that our ligand binding data is assaying the detection of IGF-1 and RGD binding to the surface of GCNT2 OE and control melanoma cells and not an assessment for both affinity and receptor number/clustering. To this end, we more accurately describe these effects as altered ligand binding “activity” to the surface of GCNT2 OE and control melanoma cells. Our data suggest that IGF-1 and RGD binding activity may be affected by altered receptor affinity or clustering, which we have not explicitly assessed in this manuscript. We have, however, comprehensively assessed differences in cell surface expression of IGF1R and various integrin subunits and we do not detect significant differences between GCNT2 OE and control melanoma cells (**Supplemental Fig. 6a-p and Figure B above**), suggesting that differences in ligand binding activity are likely not due to altered receptor number. Further studies on receptor affinity and receptor clustering are needed to clearly demonstrate these possibilities in altered ligand-binding. Discussion of these results and potential implications can be found on **Page 14 lines 360-374 and Pages 19-20 lines 510-522.**

Point iii: *There are 14 different mammalian galectins and only two were examined. Other galectins may be involved such as Galectin-9, but this is not explored.*

Our response. We agree that 14 galectins that can be examined in terms of the effects of GCNT2 expression on galectin-binding activity. However, to stay within the framework of the melanoma glycobiology inquiry, we felt that focusing on the two most highly expressed (**Supplemental Fig. 7a**) and widely-studied galectins in melanoma biology field (Gal-1 and Gal-3) would satisfy the readers.

Point iv: *Galectin binding also regulates clustering of receptors in the plane of the membrane to affect signaling, a mechanism distinct from regulation of endocytosis. This mechanism is particularly relevant for integrin signaling, yet this was not examined as requested.*

Our response. We understand that galectins can play a key role in regulating receptor clustering on the cell surface. As such, while GCNT2 overexpression inhibited galectin-3 (Gal-3)-binding (**Supplemental Fig 7d-g**), when we cultured cells with lactose, a competitive inhibitor of β -galactoside-binding galectins, including Gal-3, we did not observe any galectin-dependent IGF1R- or integrin-ECM-mediated signaling differences in melanoma cells expressing i-linear or I-branched polyLacNAc glycans (**Supplemental Fig. 7h-k and further discussed in response to Point v below**). While this finding is paradoxical to the role of galectins in helping to mediate cellular signaling events, we do believe that differential Gal-3-binding is playing a significant role in other areas of melanoma malignancy, which will be explored in future studies but are outside the scope of this manuscript.

Point v: *Adding exogenous galectin is a poor approach as it does not consider the presence of endogenous galectins. Moreover, only a single dose was examined, which may be supra-physiological and induce changes not representative. A titration is necessary, as is knockdown of endogenous galectin or use of lactose in the absence of exogenous galectin (excess exogenous galectin may prevent disruption of endogenous galectins by lactose).*

Our Response. We understand the reviewer's concern about the potential role of endogenous galectins in mediating differential IGF1R and FAK signaling, especially because melanoma cells have high expression of galectin-1 (Gal-1) and Gal-3 (**Supplemental Fig. 7a**). Additionally, while there was no difference in Gal-1-binding (**Supplemental Fig. 7b,c**) between GCNT2 cell variants, there were differences in both exogenous and endogenous Gal-3-binding between GCNT2 KD and GCNT2 OE and control melanoma cells (**Supplemental Fig. 7d,e and Supplemental Fig. 7f,g, top histograms**). Therefore, to investigate if differences in endogenous Gal-3 binding might be responsible for the observed differential IGF1R and FAK signaling in GCNT2 KD and GCNT2 OE and control melanoma cells, we incubated GCNT2 cell variants for 24hrs with 50mM lactose, a competitive inhibitor of Gal-3 or with control sugar, sucrose. We found that, when cells were incubated in lactose, Gal-3-binding was normalized (**Supplemental Fig. 7f,g, middle and bottom histograms**). Next, we stimulated control-, sucrose- or lactose-treated cells with rhIGF1 or on ECM and assayed for signaling of IGF1R and FAK. We found that the presence of GCNT2/I-branched glycans significantly inhibited phosphorylation of IGF1R and FAK under all conditions (**Supplemental Fig. 7h-k**). Overall, these data suggest that endogenous Gal-3-binding differences are not mediating differential IGF1R or FAK signaling in GCNT2 KD and GCNT2 OE and control melanoma cells. However, we are exploring other areas of malignancy in which Gal-3 binding differences may play a role but are outside the framework of this manuscript.

Point vi: *The data in Figure C have several technical issues that need to be addressed before conclusions can be drawn. Critical controls are not done side-by-side. For example, in Figure C a,b,e,f, addition of IGF1 alone needs to be compared directly to IGF1+galectin-3 in both vector control and the GNCNT2 OE and KD. Second, the endocytosis assay employed in Figure C b,f is not described and there are no error bars. Third, as noted above, experiments need to be done with triplicate staining and presented as mean with error bars.*

Our Response. All experimental controls (no treatment; rhIGF-1 alone; rhIGF1+rhGal3; rhIGF1+rhGal3+Lactose; rhGal3 alone; and rhGal3+Lactose) presented in **Figure C of the first**

rebuttal letter were assayed side-by-side at the same time and were presented in separate panels for clarity (so as to not have too many histograms in one plot). Because there were no Gal-3-dependent differences in IGF1R or FAK signaling, we did not believe it was necessary to further repeat internalization assays in the context of Gal-3. However, we repeated IGF1R internalization using rhIGF-1 alone and did not observe any significant differences in internalization of IGF1R between GCNT2 KD and GCNT2 OE and control melanoma cells (**Fig. C, panels a-d**).

Figure C. IGF1R internalization is not significantly different in GCNT2 cell variants upon stimulation with rhIGF-1. (a) Representative histograms of IGF1R cell surface expression in A375 control and GCNT2 OE cell variants with no treatment or treated with 100ng/ml rhIGF-1 for 60mins at 37°C to induce receptor internalization. Following incubation, cells were immediately placed on ice to stop internalization, washed and stained with IGF1R antibody (Biolegend) followed by 7AAD staining for dead cells. (b) Quantitation of percentage of IGF1R internalized upon stimulation with rhIGF-1 in A375 control and GCNT2 OE cell variants. (c) Representative histograms of IGF1R cell surface expression in UACC62 control and GCNT2 KD cell variants with no treatment or treated with 100ng/ml rhIGF-1 for 60mins at 37°C to induce receptor internalization. Following incubation, cells were immediately placed on ice to stop internalization, washed and stained with IGF1R antibody (Biolegend) followed by 7AAD staining for dead cells. (d) Quantitation of percentage of IGF1R internalized upon stimulation with rhIGF-1 in UACC62 control and GCNT2 KD cell variants. Results are representative of n = 3 experiments (mean ± SEM); Percent IGF1R internalized calculated by dividing MFI of IGF1R in rhIGF-1 treated cells / MFI unstimulated cells (no internalization). Statistical analyses were performed using Prism 7.0 software (GraphPad). For tests involving two groups, testing was carried out using unpaired two-tailed Student's *t* test. When more than two groups were compared, a one-way analysis of variance (ANOVA) followed by Dunnett's multiple comparisons tests were performed.

Major Point 3: *The authors were asked to demonstrate that the very small effects on IGF1R and integrin signaling were relevant to the tumor phenotype by reversal. This was not done. Given the small changes in signaling, it is unlikely that these are critical pathways responsible for the tumor phenotype yet this is the claim made in the abstract and elsewhere. The data simply do not support this conclusion.*

Our response. We understand the reviewer's comment on the implication of our work. We believe that I-branch modifications to N-glycans on many different glycoproteins by GCNT2 leads to a global change in cell surface glycosylation, which altogether significantly alters melanoma cell growth and survival. Accordingly, we have revised our Abstract and conclusions in the manuscript to better reflect the impact of our results and not imply that modifying IGF1R/integrin glycosylation

alone can regulate melanoma progression. We acknowledge that analyzed effects to these specific receptors, of which we rationalized as representative N-glycan-bearing receptors capable of controlling growth and signaling, were used as molecular surrogates for how changes in N-glycan structure can regulate different malignancy-associated pathways to alter melanoma cell growth and survival.

Therefore, in the context of this N-glycomic assessment of melanocytic/melanoma cells, we feel that additional enforcing/silencing expression of IGF1R or a particular integrin (or related signaling factor) to reverse the tumor phenotype is outside the context of this paper and is beyond the scope of a single publishable unit. This is an N-glycomic/glycozyme assessment that led to the identification of GCNT2-synthesized I-branches as major glycan modifications in melanocytic-lineage cells that leads to the malignant phenotype. This report was not an assessment to unravel which growth factor RTK or integrin receptor is most responsible for melanoma malignancy. It was a global assessment on expression of I-branched glycans on cell surface receptors. We rationalized the study of IGF1R and fibronectin-binding integrins among other family members, to serve as molecular models to study the effects of GCNT2/I-branch expression. We believe that GCNT2 modifies N-glycans on *multiple* cell surface glycoproteins and, though we focus on two major signaling pathways and related glycoprotein receptors, our results indicate that *global* cellular differences in GCNT2/I-branched glycan expression lead to a significant decrease in melanoma cell growth *in vivo*. We have now modified text in the **Abstract**, **Introduction**, **Results** and **Discussion** to better represent the global glycomic data. Moreover, we have modified our model in **Fig. 8b** to reflect the likely presence of other glycoproteins modified by GCNT2/I-branches, which may also affect *in vivo* tumor growth.

Reviewers' Comments:

Reviewer #1:

Remarks to the Author:

The points raised in the previous round of review have been satisfactorily addressed. Manuscript is suitable for publication.

POINT-BY-POINT RESPONSES TO REVIEWERS

Reviewer #1

The reviewer states, "The points raised in the previous round of review have been satisfactorily addressed. Manuscript is suitable for publication."

Our Response: Thank you! We acknowledge that all changes are satisfactorily addressed.